# WRF-Chem Simulations of Snow nitrate and other Physicochemical Properties in Northern China

Xia Wang[1], Tao Che[2,3], Xueyin Ruan[1], Shanna Yue[2,3], Jing Wang[2,3], Chun Zhao[1,4,5*], Lei Geng[1,4,5*]

[1]School of Earth and Space Sciences, University of Science and Technology of China, Hefei 230026, Anhui, China

[2]Key Laboratory of Remote Sensing of Gansu Province, Heihe Remote Sensing Experimental Research Station, Northwest Institute of Eco-Environment and Resources, Chinese Academy of Sciences, Lanzhou 730000, China

[3]College of Resources and Environment, University of Chinese Academy of Sciences, Beijing 100049, China

[4]Deep Space Exploration Laboratory, Hefei 230088, Anhui, China.

[5]CAS Center for Excellence in Comparative Planetology, University of Science and Technology of China, Hefei 230026, Anhui, China

*Correspondence to*: Lei Geng (genglei@ustc.edu.cn) and/or Chun Zhao (chunzhao@ustc.edu.cn)

**Abstract.** Snow is a key component of the cryosphere and has significant impacts on surface energy balance, hydrology, atmospheric circulation, and etc. In addition, numerous studies have indicated that snow impurities, especially nitrate, are sensitive to sunlight and can be photolyzed to emit reactive species including $NO_2$ and HONO, which serve as precursors of $O_3$ and radicals and disturb the overlying atmospheric chemistry. This makes snow a reservoir of reactive species, and this reservoir is particularly important in remote and pristine regions with limited anthropogenic emissions. The magnitude of snow chemical emissions is also influenced by snow physical properties, including snow depth, density and concentrations of light-absorbing impurities (e.g., BC and dust). Exploring and elucidating the emissions and atmospheric consequences of the snow-sourced reactive species require a global or regional model with a snow module. Here, we parameterized atmospheric nitrate deposition and its distributions in snow using a regional chemical transport model, i.e., the WRF-Chem (Weather Research and Forecasting Model coupled with Chemistry) model, and evaluated the performance of the WRF-Chem model in simulating snow cover, snow depth, and BC, dust and nitrate concentrations with field observations in northern China which is one of the regions with dense and prolong snow cover. In general, the model simulated spatial variability of nitrate mass concentrations in the top snow layer (hereafter NITS) are consistent with observations. Simulated NITS values in Northeast China from

December 2017 to March 2018 had a maximum range of 7.11–16.58 µg g⁻¹, minimum range of 0.06–

0.21 µg g⁻¹, and a four-month average of 2.72 ± 1.34 µg g⁻¹. In comparison, observed values showed a

maximum range of 9.35–33.43 µg g⁻¹, minimum range of 0.09–0.51 µg g⁻¹, and an average of 3.74 ±

5.42 µg g⁻¹. The model results show an underestimation especially in regions closes to large cities in

northeastern China, most likely due to the underestimation of $NO_x$ emissions in these regions.

Additionally, nitrate deposition, snowpack accumulation processes, and challenges in capturing fine-

scale emission variability may also contribute to the bias. These results illustrate the ability of WRF-

Chem in simulating snow properties including concentrations of reservoir species in northern China,

and in the future, we will incorporate snow nitrate photolysis in the model, exploring the emissions of

snow $NO_x$ from nitrate photolysis and the impacts on local to regional atmospheric chemistry and air

pollutant transformations.

**1 Introduction**

Through the effects on surface albedo and energy balance, snow cover has important impact on

Earth's climate system (Flanner et al., 2011). In particular, the snow's depth, grain size, and impurities

can affect its albedo, which in turn significantly influences surface warming as a result of the swift

feedback on snow structure, snow sublimation rates, and snow melt rates (He et al., 2018; Picard et al.,

2012). What is more, snow is also significant in atmospheric chemistry. Under appropriate conditions

(e.g., illuminated by sunlight), photolysis of snow impurities can lead to the release of reactive species,

including nitrogen oxides (NO + $NO_2$ = $NO_x$) and reactive halogens, to the overlying atmosphere,

disturbing atmospheric chemistry (Dominé and Shepson, 2002; Grannas et al., 2007; Zatko et al., 2016).

Improving the understanding of the spatiotemporal variations in snow physical and chemical properties

is thus important for assessing the effects of snow cover on climate and atmosphere environment.

As one of the major chemicals in snow, nitrate is perhaps one of the most important reactive species.

In particular, snow nitrate is active under sunlight, and its photolysis results in emissions of $NO_x$ and also

HONO into the boundary atmosphere (Chu and Anastasio, 2003; Zatko et al., 2013; Chen et al., 2019;

Barbero et al., 2021). In pristine regions with snow cover, long-range transported atmospheric nitrate

deposited and preserved in snow could serve as a potentially important secondary source of $NO_x$, which

is important for local production of $O_3$ and OH radicals (Bouwman, 1998; Hall and Matson, 1999; Li et

al., 2015; Logan, 1983; Nelson et al., 2023). The latter are important for atmospheric reactivity, or more

specifically, the so-called atmospheric oxidation capacity. Previous studies have investigated the photolysis of snow nitrate in polar regions. For example, when considering the emissions of $NO_x$ from snow nitrate photolysis, there is a potential doubling of $O_3$ levels and significant increases of 5 to 6 times in the level of OH in the overlying atmosphere in Greenland and Antarctica (Zatko et al., 2016). In addition to polar regions, Zatko et al. (2016b) investigated the effects of snowpack emissions on local atmospheric chemistry in a north American site located at midlatitudes but with extensive snowfall in winter. They also found significant emissions of $NO_x$ from snow, although the contribution to the local $NO_x$ budget was relatively small given that extensive anthropogenic emissions persist at the studied site, originating from traffic related to oil and natural gas extraction activities. At another mid-latitude site, Kalamazoo, Michigan, Chen et al. (2019), observed up to 44% higher HONO levels in a snowy urban environment than in other regions without snow. The high HONO concentration is in part due to snow nitrate photolysis, and the subsequent photolysis of HONO produces OH, which serves as another important OH source in addition to the common $O_3$ photolysis channel.

Northern China is renowned as one of the regions with the densest winter snow cover, boasting a broad extent of snow coverage that can reach 85% (Zou et al., 2022). Moreover, the duration of snow cover is notably extensive, with certain frigid areas experiencing snow coverage for as long as 120-140 days per year (Wang and Chen, 2022). Compared to those in polar regions, the snow in this region has higher nitrate concentrations, i.e., 0.1 - 30 $\mu g\ g^{-1}$, ~ three orders of magnitude higher than those in polar regions (0.1 - 200 $ng\ g^{-1}$) (An et al., 2022; Legrand and Mayewski, 1997; Wendl et al., 2015; Zhang et al., 2013; Jiang et al., 2021) and can receive more actinic fluxes. These conditions may facilitate snow nitrate photolysis, making it a potentially important source of $NO_x$ and HONO, which are limited from other sources in winter due to low human and bacterial activities in snow-covered regions. In addition, the prevailing northern wind in winter would transport snow-sourced $NO_x$ and/or the subsequent enhanced atmospheric $O_3$ and other species to downwind regions, including the northern China Plain (NCP), where severe haze and $O_3$ pollution occur frequently in winter. However, whether snow emission can influence atmospheric chemistry and air quality in downwind regions remains to be further investigated.

Moreover, quantifying the snow emissions of reactive species and the impacts on the overlying atmosphere as well as downwind regions requires an atmospheric chemical transport model with snow modules that can simulate snow physical and chemical properties, including snow cover, snow depth and

snow impurities (e.g., BC, dust, and nitrate concentrations), and also be able to calculate the radiative transfer of actinic flux in snow, the associated snow photochemistry and the emissions of reactive species into the overlying atmosphere. Zatko et al. (2016) have added a $NO_3^-$ photolysis parameterization to a global chemical transport model, that is the Goddard Earth Observing System (GEOS) Chemistry model (GEOS-Chem). However, as an offline model, GEOS-Chem uses archived meteorological fields, which generally have coarse resolutions, and errors can be caused by regional or smaller-scale simulations (Yu et al., 2018); moreover, GEOS-Chem cannot simulate meteorological-chemical interactions, which may be important for modeling or forecasting snow cover changes and their impacts on local to regional climate and atmospheric chemistry. Therefore, we plan to utilize the WRF-Chem model, which is an advanced, on-line regional chemistry model with a relatively well-developed snow module. However, before doing that, we have to first incorporate snow nitrate simulations into the model, which is currently not included in the snow module, and evaluate the ability of WRF-Chem in simulations of snow physicochemical properties in northern China. Therefore, this study serves as an evaluation on the performance of WRF-Chem simulations of snow coverage and snow physicochemical properties in northern China, with a development on the modeling of snow nitrate concentrations. This is the first step to use the model to investigate the effects of snow cover on local to regional atmospheric chemistry.

## 2 Model description and parameterizations

### 2.1 WRF-Chem

We use the version (v3.5.1) of WRF-Chem updated by the University of Science and Technology (USTC) of China in this study. Unlike the version distributed by NCAR to the public, the USTC version includes supplemental functionality, such as the online diagnosis of aerosol-specific radiative forcing and the aerosol-snow albedo effect (Zhao et al., 2014; Zhao et al., 2013a; Du et al., 2020). The aerosol scheme employed is the Model for Simulating Aerosol Interactions and Chemistry (MOSAIC – 8 bins) (Zaveri et al., 2008), and the gas-phase chemistry mechanism is the Carbon Bond Mechanism Z (CBM-Z) (Zaveri and Peters, 1999), both of which are used in this iteration of WRF-Chem. The MOSAIC scheme represents aerosol size distributions using eight discrete bins, with each bin covering a specific range of dry aerosol diameters: 0.039–0.078 μm, 0.078–0.156 μm, 0.156–0.312 μm, 0.312–0.625 μm, 0.625–1.25 μm, 1.25–2.5 μm, 2.5–5.0 μm, and 5.0–10.0 μm. The model considers main aerosol components including nitrate ($NO_3^-$), sulfate ($SO_4^{2-}$), chloride ($Cl^-$) ammonium ($NH_4^+$), black carbon (BC), dust, and

sea salt, and mineral dust. We note the MOSAIC aerosol scheme used in this study does not include

secondary organic aerosols (SOA), which may affect the production and phase-partition of particulate

nitrate. When simulate the generation and growth of aerosols, the MOSAIC mechanism takes into

account a variety of chemical and physical processes, including but not limited to gas-to-particle

conversion, particle nucleation, coagulation, condensation, and evaporation. Additionally, the model

considers important processes of aerosol deposition, including both dry and wet deposition. These

processes are critical for comprehending the behavior and fate of aerosols, including their incorporation

in snow. In the model, particle diffusion and gravitational effects are considered to simulate dry

deposition of aerosol (Binkowski and Shankar, 1995). Wet deposition, including rainout, washout and

scavenging processes, is also simulated in the model to accurately represent the removal of aerosols

through precipitation following the methodologies outlined by Easter et al. (2004) and Chapman et al.

(2008). This research does not explicitly model cloud-ice-borne aerosols. However, it does consider the

elimination of aerosols as they undergo freezing within droplets. The removal of aerosols by convection

transport and their wet deposition via cumulus clouds are modeled according to the methods described

by Zhao et al. (2013b). This study utilizes the Community Land Model (CLM) v4.0 (Lawrence et al.,

2011) coupled with the Snow, Ice, and Aerosol Radiative Model (SNICAR) (Flanner and Zender, 2005)

as an option for the land surface model (Jin and Wen, 2012).




**Table 1.** An overview of the model configurations utilized.

| Option | Parameterization schemes |
|---|---|
| Simulation periods | October 2017 to March 2018 |
| Horizontal resolution | 36 km |
| Model spin-up time | 2 months |
| Vertical levels | 41 (About 8 layers beneath the surface of 1 km) |
| Domain sizes | 149 × 189 |
| Photolysis scheme | Fast - J |
| Aerosol chemistry | MOSAIC 8 bin |
| Gas-phase chemistry | CBM-Z |
| Land surface scheme | CLM land surface scheme |
| Microphysics | Morrison 2-moment |
| Longwave Radiation | RRTMG |
| Shortwave Radiation | RRTMG |
| Planetary boundary layer | YSU |
| Cumulus Cloud | Kain–Fritsch |

**2.2 Snow simulations in WRF-Chem**

Snow accumulations on land surface as well as the physicochemical properties are calculated using SNICAR model in WRF-Chem. This mode incorporates a layered structure, considering the vertical variability of snow properties and accounting for the heating effects of the underlying ground and its influence on snow characteristics (Flanner et al., 2012; Flanner and Zender, 2005; Flanner et al., 2009;

Flanner et al., 2007). The theoretical framework of Wiscombe and Warren (1980) and the two-stream, multi-layer radiative scheme proposed by Toon et al. (1989) are employed within SNICAR. It has excellent performance in simulating snow surface albedo, radiative absorption within snow layers, snow impurities, and radiative effects within snow. It was initially utilized by Flanner et al. (2007) to investigate snow aging and aerosol heating in a global climate model. The simulated changes in snow albedo based

on specific black carbon (BC) concentrations have been validated through field measurements and laboratory experiments (Brandt et al., 2011; Hadley and Kirchstetter, 2012). In CLM, there are five thermal layers that correspond to the radiative layers defined by SNICAR, enabling the vertical resolution of densification, snow meltwater transport, and thermal processes (Oleson et al., 2010a). For a more

comprehensive understanding of the SNICAR model, refer to Flanner and Zender (2005) and Flanner et

al. (2012); (Flanner et al., 2007).

To simulate snow nitrate photolysis and its impacts on overlying atmospheric chemistry, one needs to obtain snow cover, snow depth, and snow physical and chemical properties, including snow density; impurities, including BC, dust; and nitrate. Physical properties are used to simulate radiative transfer in snow. While nitrate and other impurities in snow also influence radiative transfer snow, and especially nitrate in snow is the source of snow-sourced $NO_x$. Currently, all other components (e.g., BC, dust) but not nitrate have been included in SNICAR and parameterized by Zhao et al. (2014). In this study, we parameterized and included snow nitrate concentration in simulation.

**2.2.1 Parameterization of nitrate concentrations in snow**

Currently, SNICAR does not include calculations of nitrate concentrations embedded in snow. In principle, concentrations of nitrate within each snow layer are mainly influenced by atmospheric deposition flux and snow accumulations. After deposition, layer combinations and divisions, and, in a rare case, meltwater flushing may also take effect. To quantify nitrates in snow, in this study, we parameterized nitrate concentrations in snow by considering the deposition processes of nitrate, including both dry and wet deposition. Dry deposition processes (sedimentation and turbulent mix-out) directly contribute to the accumulation of particulate and gaseous nitrate in surface snow. For gaseous nitrate ($HNO_3$), the dry deposition flux (kg m$^{-2}$ s$^{-1}$) is calculated using the following equation:

$$F_{dry\_gas} = V_{HNO3} \times C_{HNO3} \times D_{air} \qquad (1)$$

where $V_{HNO3}$ is the dry deposition velocity of gaseous nitrate (m s$^{-1}$), $C_{HNO3}$ is the concentration of gaseous nitrate in the first (i.e., surface) layer of the atmosphere (ppmv), and $D_{air}$ is the air density in the surface layer (kg m$^{-3}$). For particulate nitrate, the dry deposition flux is calculated for each aerosol size bin as follows:

$$F_{dry\_aer} = \sum_i (V_{no3a\_i} \times C_{no3a\_i} \times D_{air}) \qquad (2)$$

where $V_{no3a\_i}$ is the dry deposition velocity of particulate nitrate in each size bin (m s$^{-1}$), $C_{no3a\_i}$ is the concentration of particulate nitrate in each size bin (μg kg-dryair$^{-1}$), and $D_{air}$ is the air density in the surface layer (kg m$^{-3}$).

For wet deposition, it includes both in-cloud and below-cloud scavenging of gaseous nitrate and particulate nitrate including cloud-borne nitrate. In-cloud scavenging refers to the incorporation of aerosols and gases into cloud droplets as they form within clouds. Below-cloud scavenging (washout) refers to the removal of particulate and gaseous nitrate by falling hydrometeors as they descend below the cloud, where nitrate compounds are captured through mechanisms like Brownian motion, electrostatic forces, collision, and impaction, ultimately leading to their deposition on the snow surface. In this study, we estimate the amount of nitrate wet deposition by calculating the concentration changes of atmospheric total nitrate during in-cloud and below-cloud scavenging processes. For in-cloud scavenging, the concentration of cloud-borne nitrate and gaseous nitrate removed is based on the following equation:

$$\Delta C_{in-cloud} = C_{no3-cw} \times Scale_{in,cw} + C_{HNO3} \times Scale_{in,gas} \tag{3}$$

where $C_{no3-cw}$ is the concentration of cloud-borne nitrate aerosols in cloud, $C_{HNO3}$ is the concentration of gaseous HNO$_3$ in cloud, and $Scale_{in,cw}$ and $Scale_{in,gas}$ represent the scaling factors for in-cloud scavenging that indicate the amount of nitrate removed in cloud, respectively.

For below-cloud scavenging, the removal of nitrate aerosols and gases is represented as:

$$\Delta C_{below-cloud,i} = C_{no3a,i} \times Scale_{below,aer,i} + C_{HNO3,i} \times Scale_{below,gas,i} \tag{4}$$

where $C_{no3a,i}$ is the concentration of nitrate aerosols below the cloud in layer $i$, and $Scale_{below,aer,i}$ and $Scale_{below,gas,i}$ represent the factors for below-cloud scavenging that indicate the amount of nitrate removed by impaction-interception in each atmospheric layer. In this study, the calculations of the scavenging scales for both in-cloud and below-cloud wet removal of nitrate aerosols and gases are based on the methodologies of Easter et al. (2004) and Chapman et al. (2008).

The total nitrate concentration used for wet deposition calculations is the sum of the concentrations removed during scavenging.

The wet deposition flux (kg m$^{-2}$ s$^{-1}$) is then calculated using the following equation:

$$F_{wet} = \Delta C_{in-cloud} + \sum_{i}^{n} \frac{\left(\Delta C_{below-cloud,i} \times D_{air,i} \times H_{air,i}\right)}{dt} \tag{5}$$

where $D_{air,i}$ is the air density in each atmospheric layer. $H_{air,i}$ is the thickness of the atmospheric layers (m), and $dt$ is the model time step (s). The variable $n$ represents the number of atmospheric layers below the cloud.

After deposition, nitrate is mixed instantly and uniformly in the model surface layer, which never
exceeds 3 cm thick. The nitrate mass concentration in surface snow ($M_{\text{NITS}}$: kg kg$^{-1}$) was calculated by
deposition fluxes of atmospheric nitrate as follows:

$$M_{NITS} = \frac{\left( \left( F_{dry} + F_{wet} \right) \times dtime \right)}{W_{sno}} \tag{6}$$

where *dtime* is the land model time step used in SNICAR(s), as distinct from the dt mentioned above
used in atmospheric processes, and $W_{sno}$ is the snow mass in the surface layer (kg m$^{-2}$). Furthermore,
the CLM continuously builds a new surface snow layer when a fresh snowfall event occurs, and nitrate
mass concentrations in surface snow are updated as follows:

$$M_{NITS}^{new} = M_{NITS} + \frac{\Delta F \times dtime}{\Delta W_{sno}} \tag{7}$$

where $\Delta F$ is the cumulative wet and dry deposition of atmospheric nitrate during the entire period
between the newly fallen snow and the previous time step, *ΔWsno* is the newly gained snow mass
during the entire period between the newly fallen snow and the previous time step, and *Δt* is the period
spanning from the newly fallen snow to the previous time step.

By repeating the above motioned processes, a snowpack with initial nitrate concentrations in each
layer of the snowpack was simulated.

**2.2.2 Potential modification by melting processing after deposition**

Previous studies have shown that at midlatitudes, snow melt occurs occasionally, which will modify
the concentrations of impurities (Zhao et al., 2014; Flanner et al., 2007; Eichler et al., 2001). Following
similar processes, we considered the potential effects of these processes on snow nitrate concentrations.
In particular, the melting of snow can redistribute nitrate (and other species) through the introduction of
excess water into the layer beneath when the meltwater surpasses the layer's retention capacity, which is
determined by irreducible water saturation and snow porosity. The rate of change in nitrate mass for each
layer i, due to its incorporation into meltwater, is directly proportional to the mass mixing ratio and with
the adjustment of a scavenging factor, which can be described as follows:

$$\frac{dmi}{dt} = k(q_{i+1}c_{i+1} - q_i c_i) + D \tag{8}$$

where $m_i$ represents the total mass of nitrate within layer $i$, which is affected by the removal efficiency
*(k)* and the water flux leaving the layer *(qi)*. The concentration of nitrate in layer i, denoted as *ci*, is the
proportion of the nitrate mass to the total mass of water in both liquid and solid forms within that layer.

The term $q_{i+1}c_{i+1}$ represents the mass flux of water leaving the layer above ($i+1$) multiplied by the concentration of nitrate in that layer, accounting for the transfer of nitrate from the upper layer to the current layer. $D$ represents the combined effect of total atmospheric particulate and gaseous nitrate deposition, which is specifically added to the surface layer of the snowpack. In this study, following Flanner et al. (2012) and Zhao et al. (2014), the scavenging ratio (k) for nitrate is assumed to be 0.2. This value is highly uncertain for nitrate and needs to be constrained by future observations (Flanner et al., 2012; Qian et al., 2014; Zhao et al., 2014). However, for this process to be effectively impactful, significant melting would need to occur. During our simulation period, temperatures in northern China were consistently low, primarily below 0°C, and significant melting did not take place. Therefore, we believe the impact of this assumption is minimal in this context. It is worth noting that the portion of nitrate mass lost through meltwater from the bottom layer of snow is considered to be removed from the snowpack and is not accounted for within the model.

In summary, the nitrate concentrations in each snow layer are determined by factors such as atmospheric deposition rates, the amount of new snowfall, layer combinations and divisions, and meltwater flushing (Oleson et al., 2010b; Flanner et al., 2012; Flanner et al., 2007). When snow layers are combined or divided, nitrate masses are redistributed proportionately with snow masses conserving nitrate masses within the snow column.

## 2.3 Numerical experiments

The study employed simulations covering the entire area of China, utilizing a spatial resolution of 36 × 36 km with a grid composed of 149 × 189 cells, as depicted in Fig. S1. The simulations run from December 2017 to March 2018 covering the field campaign period, with an additional two months modeled before December 2017 as the model spin-up. The starting and side boundary conditions for meteorology are drawn from the NCEP Final reanalysis dataset, which provides data at a resolution of 1° horizontally and at 6-hour intervals. The specific model setup employed in this research is outlined in Table 1, including the Yonsei University (YSU) planetary boundary layer scheme, the Kain-Fritsch cumulus parameterization scheme, the Morrison two-moment microphysics scheme, the Rapid Radiative Transfer Model (RRTMG) for both longwave and shortwave radiation, and the Community Land Model (CLM) for land surface processes. The YSU scheme was chosen to parameterize the planetary boundary layer processes, while the Kain-Fritsch scheme addresses the representation of convective clouds. The

Morrison scheme handles microphysical processes, capturing the characteristics of cloud and
precipitation formation. The RRTMG schemes accurately modeled longwave and shortwave radiation
interactions. Finally, the CLM scheme accounted for land surface interactions. By integrating these
schemes, this study aimed to provide comprehensive simulations and insights into the atmospheric and
snow processes and interactions involved during the selected period.

For the purpose of modeling anthropogenic emissions, we utilize the 2015 version of the Multi-
resolution Emission Inventory for China (MEIC), which offers a fine resolution of $0.1° \times 0.1°$ (Li et al.,
2017a; Li et al., 2017b). To determine the vertical distribution of dust, we apply the GOCART dust
emission scheme developed by Ginoux et al. (2001). Subsequently, the generated dust particles are
assigned to numerous size categories within the MOSAIC aerosol scheme, adhering to the scale-invariant
fragmentation mechanics for brittle materials as described by Kok (2011). Additional information
regarding the integration of the dust emission scheme with the MOSAIC aerosol scheme in WRF-Chem
is available in Zhao et al. (2010). Hourly resolved biomass burning emissions, with a 1 km horizontal
resolution, are obtained from the Fire Inventory from NCAR (FINN) (Wiedinmyer et al., 2011). Biogenic
emissions were calculated using the MEGAN v2.0 model.

## 2.4 Observations

### 2.4.1 Snow and its physicochemical properties datasets

The main dataset for observing and assessing the simulations of snow and its physicochemical
properties was obtained from the field study conducted by Che et al. (2022) from December 2017 to
March 2018, when snow was collected from more than 200 road trips in both the Northeast and Northwest
regions of China. The observational route in Fig. 1 covers three distinct regions with different climatic
zones, underlying surfaces, and elevations. Extensive field observations were conducted to study snow
cover characteristics along this route. The sampling points are labeled in Fig. 1 and categorized by region
and month. At each site, snow samples were collected. A total of 269 snow samples were collected, and
the concentrations of nitrate and calcium and the snow depth were measured.

For further information regarding the measurements conducted during the Northern China campaign,
additional details are available in Che (2020). During the campaign, snow samples were collected at
various depths, yet the concentration of ions was chiefly calculated for the top layer of the snow. Hence,

we compare the simulated ion mass content within the uppermost 2 cm of the surface snow layer against the average observational data derived from snow samples collected at depths ranging from 2 to 5 cm.

## observation station sites

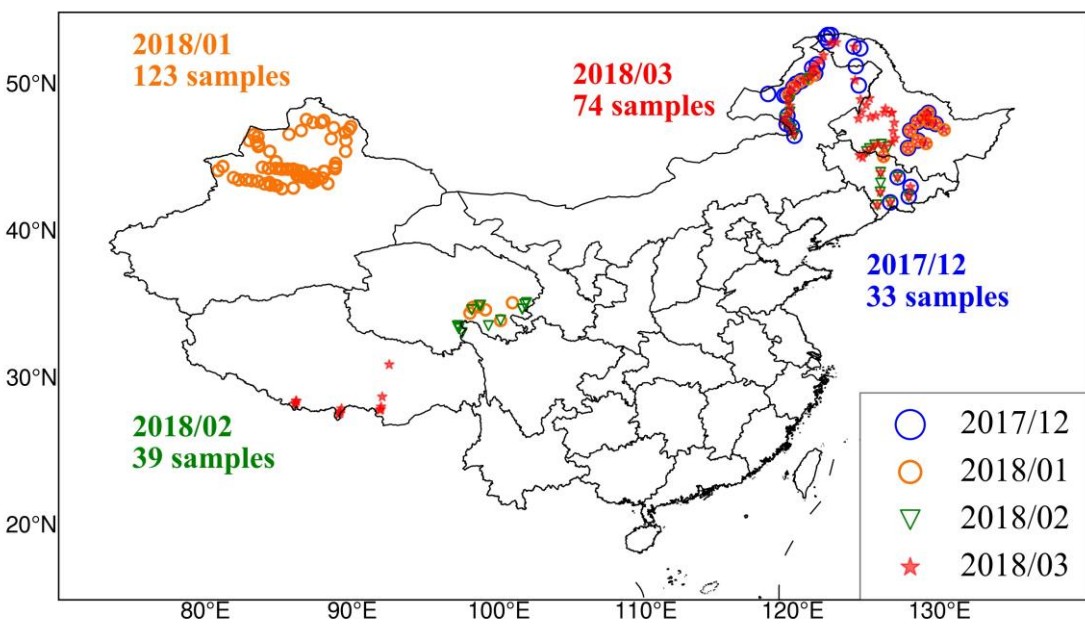

**Figure 1.** Sampling points along the road trip from December 2017 to March 2018 are marked with different colors to represent different months. Color indicates different months of the observations.

### 2.4.2 MODIS-based snow area product dataset

In addition to field observations of snow impurities and depth, we also collected snow cover data for comparison with the model simulations. For this study, we utilized the daily cloud-free 500 m snow cover dataset over China, compiled by Hao et al. (2022). This dataset is provided as a long-term time series resource, offering fine spatial detail at 500 m × 500 m. More information is available at the homepage of the National Cryosphere Desert Data Center of China (http://www.ncdc.ac.cn).

**2.4.3 Meteorological and air quality data**

To evaluate the model's performance concerning surface temperature and rainfall, which are important for snow simulation, we obtained meteorological data on temperature and precipitation from the National Climatic Data Center (NCDC) (https://www.ncdc.noaa.gov/). The NCDC has more than 400 ground stations in China, and data have been collected since 1942.

## 3 Results and Discussion

### 3.1 Meteorological simulations

Two key factors affecting the snow simulation are surface temperature and snow precipitation. Since there are no publicly available observation data for surface snow precipitation, only temperature was compared with observations for model evaluation. Figure 2 displays the 2 m temperature patterns across China simulated by WRF-Chem and observed, with the left panel showing the spatial distribution of temperature, and the right panel illustrating the scatter plot comparison between simulation and observation. The background color in the left panel represents the average simulated values from December 2017 to March 2018, while the right panel shows the daily averages from the simulation corresponding to the observation dates at each station. The scatter plot also distinguishes between regions, with orange dots representing northern China and blue dots representing southern China. Daily 2 m temperature data from December 2017 to March 2018 at 415 sites in China were sourced from the National Oceanic and Atmospheric Administration (NOAA). Based on the graph, it is evident that the model accurately depicts the spatial patterns and fluctuations in the 2 m temperature, aligning well with the observed data. Furthermore, the simulation accurately represents the notable decrease in the 2 m temperature as latitude increases, ranging from near freezing levels to approximately -30°C. From the scatter plot on the right, it can be observed that the model generally performs well in simulating the 2 m temperature, closely aligning with the observed data. However, there is a slight underestimation of temperature for southern China and a slight overestimation for northern China. Such systematic biases have also been reported in other studies (Gao, 2020; Gao et al., 2022; Kong et al., 2019; Yu et al., 2011). These discrepancies may be attributed to the complexity of regional climate factors, such as varying land surface characteristics, boundary layer processes, and the challenges of accurately simulating localized weather phenomena like cold fronts or temperature inversions in certain regions. Furthermore, differences in the representation of terrain and vegetation between the model and reality could contribute to these systematic errors, particularly in regions with complex topography (Gutowski et al., 2020).

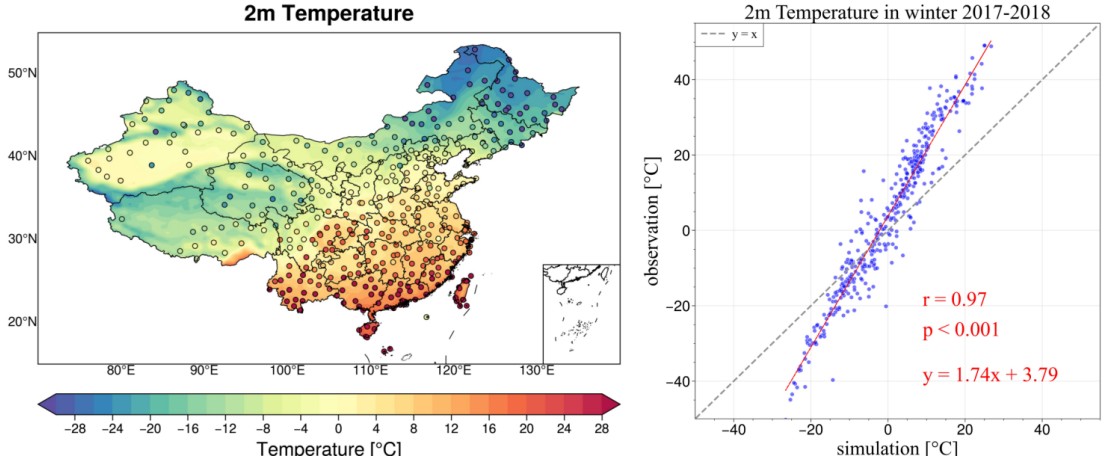

**Figure 2.** Spatial distribution of 2 m temperature observed and simulated by WRF-Chem over China averaged from December 2017 to March 2018. [Left: The background color represents the average simulated value. Right: Simulated values are the daily averages corresponding to the observation dates at each station.]

## 3.2 Snowpack simulations

### 3.2.1 Spatial distributions of snow cover

Before analyzing the patterns of light-absorbing impurities in snow, it is crucial to assess the simulated snow cover produced by WRF-Chem. Figure 3 shows the spatial patterns of snow cover (the percentage of land area with snow at each grid cell) from the WRF-Chem simulation (first column), MODIS-based observational data (second column), and the difference between the two (third column) from December 2017 to March 2018, providing the average results for each month. The MODIS data has been averaged to the 36 km WRF grid for a fairer comparison. The third column shows the difference map, calculated as the WRF-Chem simulation minus the MODIS observations, highlighting areas where the model either overestimates or underestimates snow cover. Here, snow cover is defined as the snow fraction [0-1], which represents the percentage of land area with snow at each grid point. Both simulations and observations indicate that snow cover is concentrated primarily in China's northeastern, northwestern, and Qinghai–Tibet Plateau regions. The distribution of snow cover generally follows the temperature pattern. Areas with lower temperatures tend to have greater snow cover. The highest snow cover percentage, up to 90%, is observed in the northeastern region. Both the observations and simulations reveal snow accumulations in central China in January 2018. The difference between the simulation and MODIS data in the third column reveal systematic biases. In particular, the WRF-Chem model tends to

overestimate snow cover in parts of northern China, especially in regions with complex terrain or higher altitudes. This overestimation could be attributed to the model's potential oversensitivity to cold temperatures or its overestimation of snowfall in these colder regions. Complex terrain can also challenge

the model's ability to accurately simulate microclimatic conditions, leading to discrepancies in snow cover estimates. Conversely, in southern and central China, the model underestimates snow cover, likely due to limitations in how WRF-Chem handles snow accumulation and melting in warmer areas. Overall, the model appears to reasonably capture the stable snow cover in most of the regions of interest, though some discrepancies remain related to small-scale surface features caused by terrain, with most biases

staying within 30%.

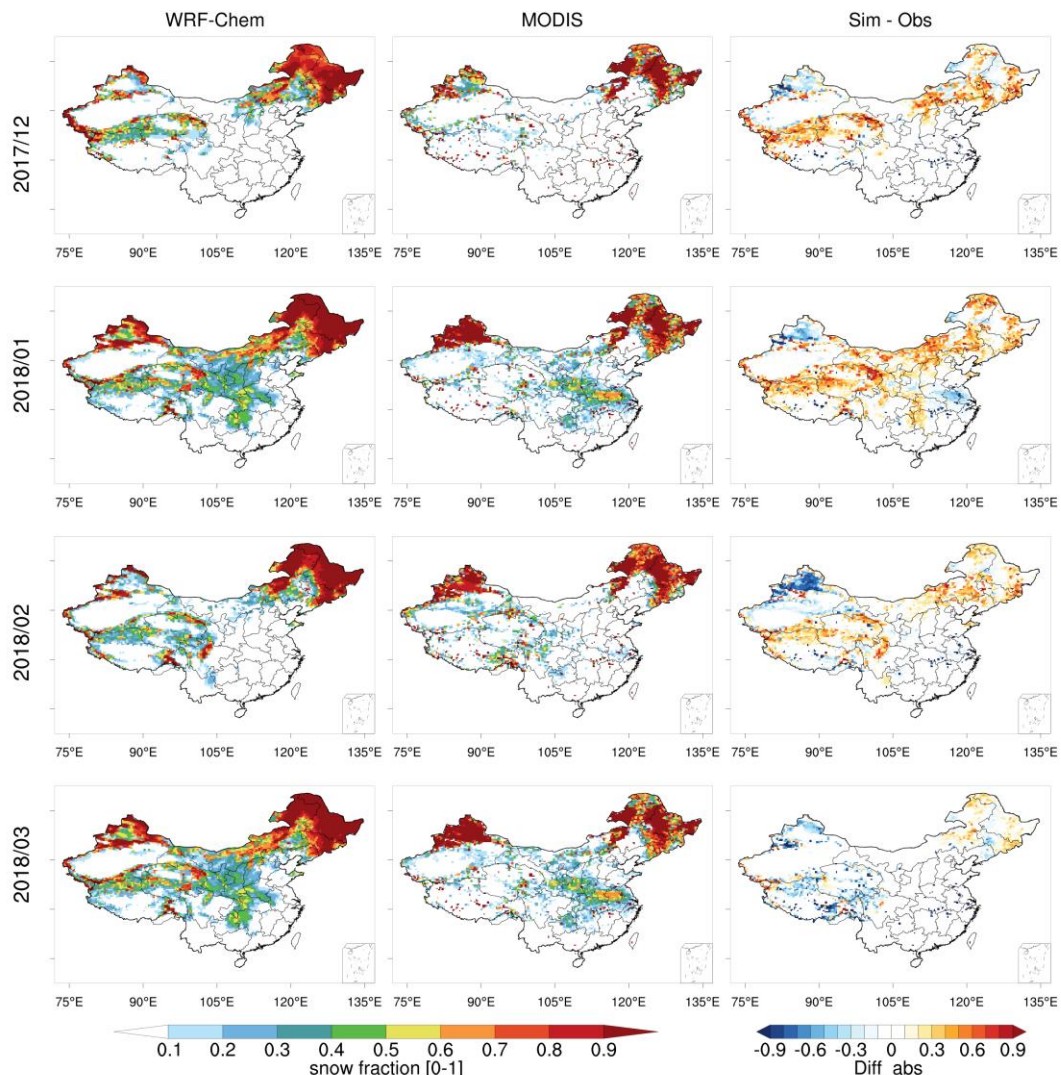

**Figure 3.** Spatial distribution of snow cover simulated by WRF-Chem and observed from MODIS-based data across China from December 2017 to March 2018. The data presented are monthly averages, and the third column shows the difference calculated as the simulation minus the observation.

### 3.2.2 Snow depth

Figure 4 illustrates the simulated snow depth from December 2017 to March 2018 along with the observations from the field campaign integrated into each panel. The background color in the figures represents the monthly average of the simulation results, while the plotted dots indicate the observed data for that month. In general, the simulation and observation results exhibit consistency in terms of spatial distributions and magnitudes, both of which indicate that snow depth is deeper in a northly direction. In particular, among the four months, January 2018 had the highest number of observations, distributed across the northeastern, northwestern, and Tibet plateau regions of China. The observed data for the other three months are primarily concentrated in the northeastern region of China. Detailed comparison maps for specific regions can also be found in Fig. 4. Over time, the snow cover in the northeastern region exhibits dynamic variations (I, III, V, and VII in Fig. 4). From December 2017 to March 2018, there was a gradual increase in snow depth each month, reaching its maximum in March 2018. In January 2018, in the northeastern region, the model well captured the spatial variations in the observed snow depths, with excellent agreement in both high- and low-value areas. (VII in Fig. 4). In March 2018, extensive observational data were collected, primarily focused on the western Greater Khingan Mountains and the Northeast China Plain in Northeast China. From the panel, we can observe that the overall simulation performance is quite satisfactory, as it effectively captures the spatial variations in snow depth. The simulation successfully reproduced high snowfall values around the Hulunbuir area, located in the Inner Mongolia Autonomous Region, reaching up to 29 cm. The March snow variations are clearly visible in the figure, with snow depths reaching more than 20 cm in both the western Greater Khingan Mountains and the Northeast China Plain. In addition, we extracted the simulated values corresponding to the observations at each station and plotted them in a scatter plot (Fig. 11a). From the results, most of the simulated snow depths align reasonably well with the observations, though underestimation is evident in some areas, particularly in regions with lower snow depths.

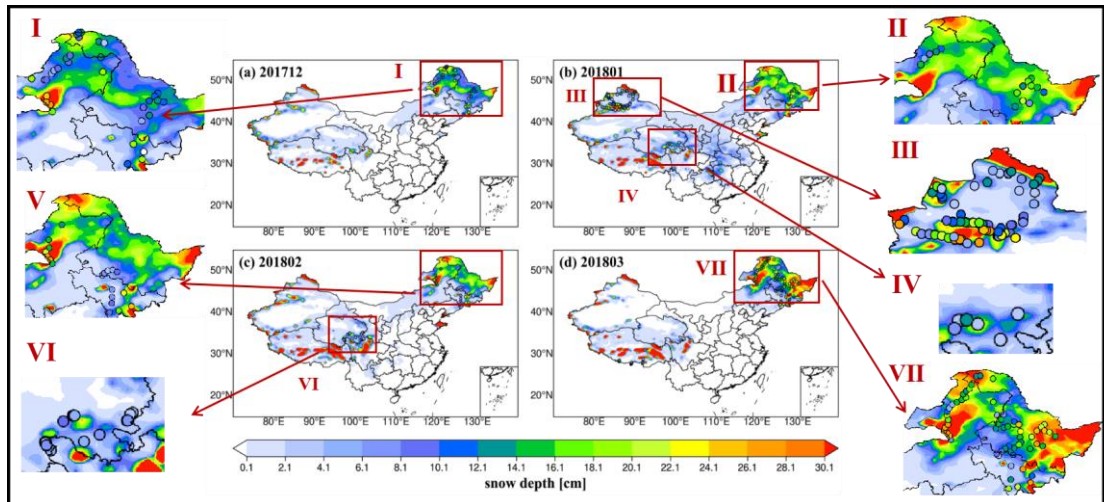

**Figure 4.** Spatial distribution of snow depth (cm) observed and simulated by WRF-Chem and across China from December 2017 to March 2018. [Note: The background color in each figure represents the monthly average of the simulation results, while all the observations for each month are embedded in each panel.]

### 3.3 Modeling of DUST and BC concentrations in snow

### 3.3.1 Spatiotemporal variability of Dust in snow

Figure 5 illustrates the pattern of dust concentration (mg/g) in the top snow layer simulated by WRF-Chem. Additionally, in the absence of direct dust measurement data, we have graphed the concentration of the dust tracer calcium ions in the top layer of snow a simulated by the WRF-Chem model together with the field campaign observations for each month (Fig. 6). From the figure, it is clear that there are significant differences between the dust distributions in each month. For example, in December 2017, the dust is mainly concentrated in the northwestern part of China, while in January 2018, it spreads to other parts of the northwest and northeast. In February 2018, the dust seems to be more evenly distributed across northern China, and in March 2018, it is mainly found in the northeastern part of China. These variations are associated with the distribution of snow cover in each month. Overall, dust is primarily distributed across Northwest China, Mongolia, and Liaoning Province, corresponding with the distribution of dust sources (see Fig. S2). In regions near dust source, DSTS is highest (> 3 mg g$^{-1}$). As far away from the source region, DSTS gradually decreases. In Northeast and Central China, the concentration decreases to approximately 10 μg g$^{-1}$, and at the further northern boundaries, the concentrations can even drop as low as ~100 ng g$^{-1}$.

The modeled calcium ion content in the snow was calculated based on the proportion of calcium carbonate in the GOCART dust emission mechanism used in WRF-Chem, where calcium is assumed to

445 constitute 0.4% of the total dust mass, and carbonate ($CO_3^{2-}$) accounts for 0.6% (Ginoux et al., 2001;

Kok et al., 2014a; Kok et al., 2014b). However, as our research area is northern China, particularly the

northwest and Loess Plateau, which is a major source area for dust, these default proportions are not

representative of the real values as suggested by observations conducted in these regions which indicates

the mass fraction of calcium in dust ranges from 7% to 12% in northern China (Zhang et al., 2003).

Therefore, we used the average observed fraction of 9.5% to calculate the modeled calcium

concentrations in this study. Field-observed calcium ion concentrations in the top layer of snow (CAS)

are indicated by dots superimposed in Fig 6. The simulated CAS values closely match the observations,

effectively capturing the spatial variation and magnitude of the CAS. The simulation shows the highest

CAS, exceeding 10 µg g$^{-1}$, across Northwest China (90–100°E, 40–50° N) in January 2018, where the

455 DSTS is also the highest (b) in Fig. 5. Moreover, the high CAS values simulated during this month align

well with the actual measurements recorded, accurately capturing the overall spatial variations. CAS and

DSTS exhibit similar distribution patterns and are primarily concentrated in the northern and

northwestern regions of China, such as Inner Mongolia and Liaoning Province. These areas are

characterized by arid and semiarid climatic conditions and desert landscapes, increasing susceptibility to

460 the dispersal of dust particles. In addition, we extracted the simulated values corresponding to the

observations at each station and plotted them in a scatter plot (Fig. 11b). From the results, the simulated

snow calcium ion concentrations generally fall within the same order of magnitude as the observations.

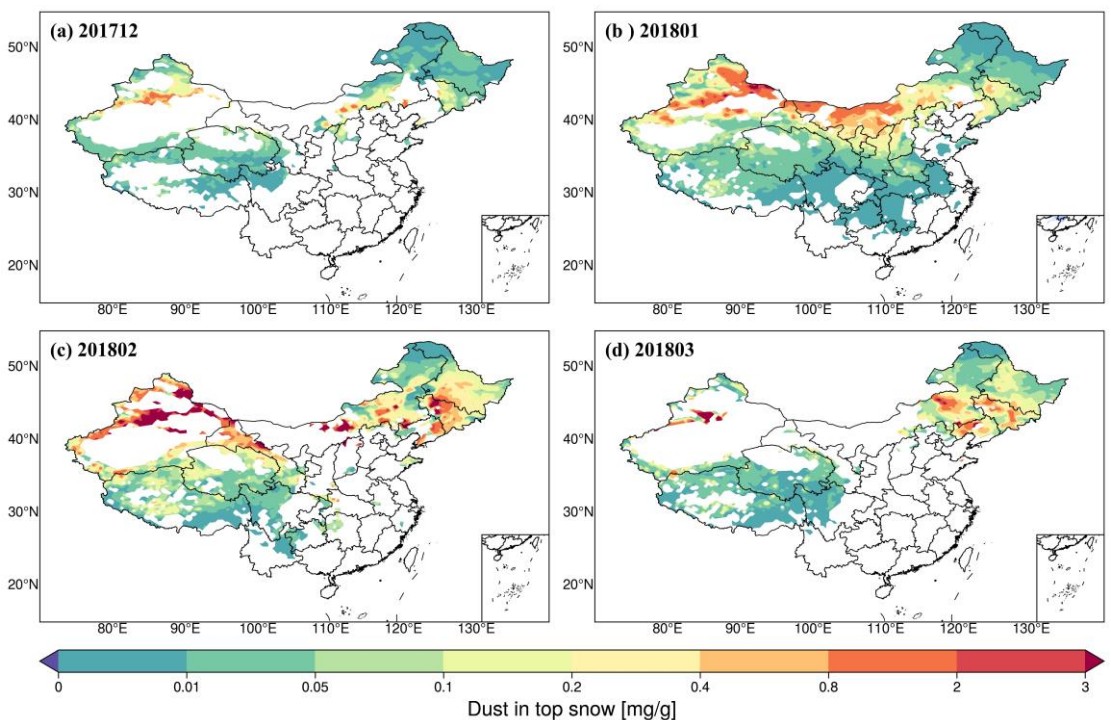

**Figure 5**. Spatial distribution of dust concentrations (mg/g) in the top snow layer simulated by WRF-Chem across China from December 2017 to March 2018 (a–d).

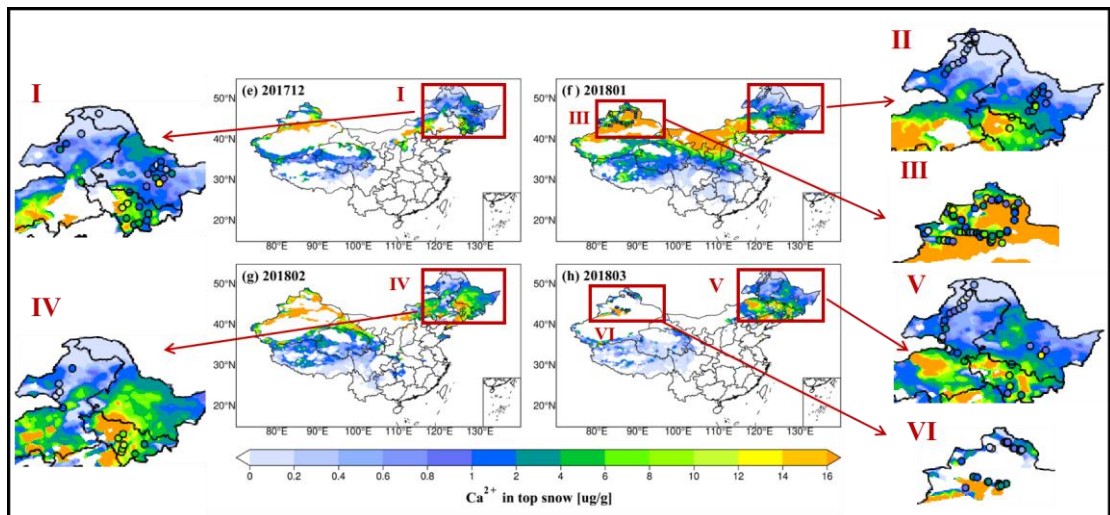

**Figure 6**. Spatial distribution of calcium ion concentrations (ug/g) in the top snow layer simulated by WRF-Chem, with field campaign observations embedded at specific locations for each month (e–h) across China from December 2017 to March 2018.

### 3.3.2 Spatiotemporal variability of BC in snow

Figure 7 displays the modeled black carbon (BC) concentration distributions across space in the top layer of snow across China from December 2017 to March 2018, simulated by WRF-Chem. We do not have BC observations during the simulation period, but there were observations in other winters for the same regions. We compared our simulations with those observations, and the results show that both the magnitude and the spatial patterns of our simulated BC concentrations are consistent with the observed values reported in the literature (Zhao et al., 2014). In addition, Zhao et al. (2014) also used the same model and framework to simulate BC concentrations during the observational period. Their study showed reasonable agreement with a median model-to-observation ratio of 1.03. In the vicinity of approximately 40° N and 125° E in Northeast China, as depicted in Fig.7, the highest concentrations of BC in the top snow layer (BCS) reach more than 6000 ng g$^{-1}$. This region is characterized by significant snow cover and depth, as illustrated in Figs. 4 and 5. As far away from northeast China, the BCS decreases and drops to less than 50 ng g$^{-1}$ towards the northwest border of China. This finding aligns with the results of Zhao et al. (2014), who reported high BCSs in areas of dense industrial activity and reduced levels (30–50 ng g$^{-1}$) at more northerly latitudes in the northern reaches of China, around 51° N. The large spatial and temporal variations in BCS are influenced in part by the changes in snow conditions (Fig. 4) and its BC

content as represented in the model. During the initial accumulation of snow, the mass of BC within the snow is significantly less than the mass of the snow itself, leading to the lowest recorded BCS. As the

snow begins to melt, BCS continues to rise primarily due to melt enrichment, where melting snow concentrates BC near the snow surface (Doherty et al., 2013). This effect is further enhanced by dry deposition until the snow completely melts. Note that in Jan. 2018, a high concentration of BC was simulated in central China, which was due to the low snow accumulation at that time (i.e., low snowfall but high BC emissions led to high snow BC content).

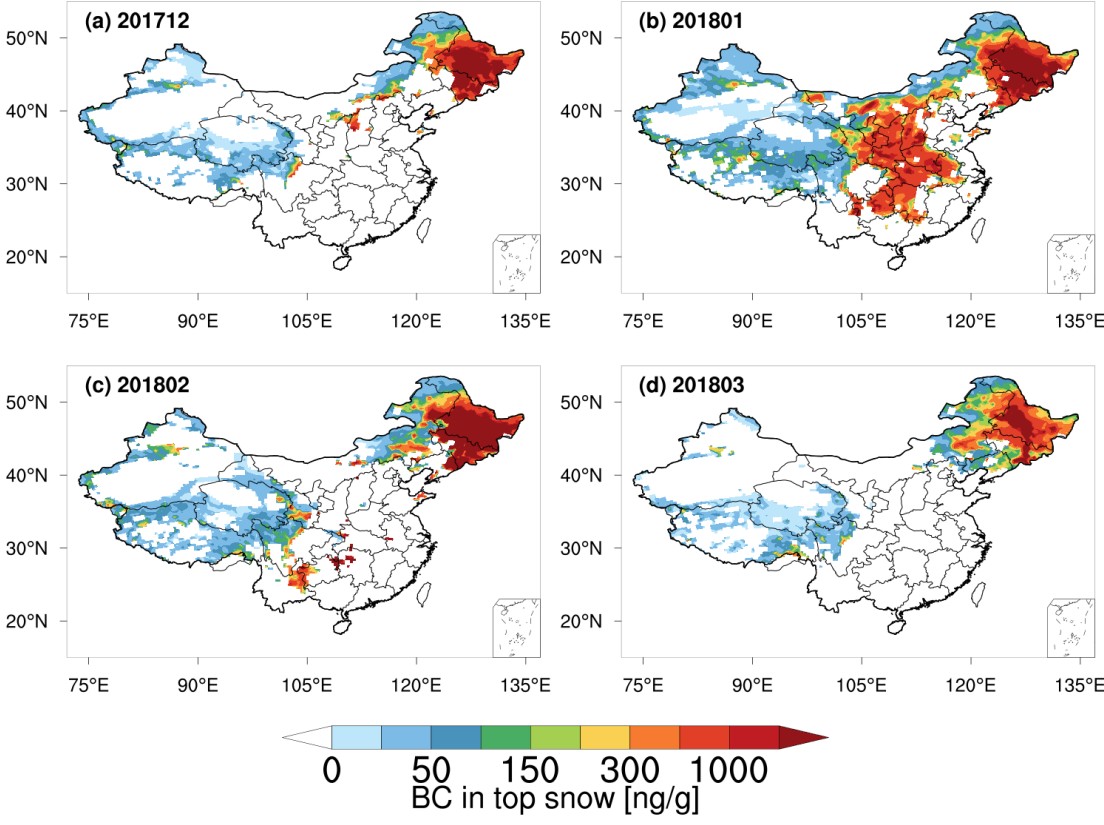

**Figure 7.** Spatial distribution of BC concentration (ng/g) in the top snow layer simulated by WRF-Chem across China from December 2017 to March 2018. [Note: The background color in each figure represents the monthly average of the simulation.]

### 3.4 Snow nitrate simulations

#### 3.4.1 Deposition fluxes of nitrate on snow

    Snow nitrate concentration was calculated by incorporating dry and wet total nitrate (atmospheric gaseous and particulate nitrate) deposition fluxes into the SNICAR module in WRF-Chem. Therefore,

before assessing the concentration of nitrate in the snowpack, it is necessary to evaluate the

reasonableness of the model-derived deposition fluxes. The simulated spatial distributions of nitrate deposition via dry and wet processes on snow from December 2017 to March 2018 are shown in Fig. 8. Deposition fluxes only accumulate across snow-covered surfaces, both in space and time. To assess nitrate deposition fluxes in winter China, we initially compared our simulation results with findings from other simulation studies (Liu et al., 2022a; Ma et al., 2023b; Zhao et al., 2015). Overall, our simulation results exhibit consistency in terms of the spatial distributions and magnitudes of atmospheric nitrate deposition. During the studied period spanning from December 2017 to March 2018, the monthly deposition flux of nitrate (including both dry and wet depositions of gaseous and particulate nitrate) in China was approximately $0.17 \pm 0.007$ (mean $\pm 1\sigma$) kg N ha$^{-1}$ month$^{-1}$. Among them, dry deposition contributed approximately $0.07 \pm 0.005$ kg N ha$^{-1}$ month$^{-1}$, while wet deposition accounted for $0.09 \pm 0.007$ kg N ha$^{-1}$ month$^{-1}$. Wet deposition comprised a slightly greater proportion, constituting 56% of the total deposition flux. In comparison, Yu et al. (2019) utilized linear regression and Kriging interpolation methods drawing upon data from the Nationwide Nitrogen Deposition Monitoring Network (NNDMN), finding that the monthly dry deposition flux of nitrate (including both gaseous and particulate nitrate) over China from 2011 to 2015 was approximately $0.27 \pm 0.08$ (mean $\pm 1\sigma$) kg N ha$^{-1}$ month$^{-1}$ and wet deposition flux was approximately $0.31 \pm 0.23$ kg N ha$^{-1}$ month$^{-1}$. Note this monthly average are values considering data from all 12 months but not only in winter. If considering winter only means, the dry and wet deposition fluxes are ($0.09 \pm 0.03$ kg N ha$^{-1}$ month$^{-1}$) and ($0.10 \pm 0.07$ kg N ha$^{-1}$ month$^{-1}$), respectively, assuming the monthly means are approximately 1/3 of summer means according to previous nitrate deposition of seasonal research findings (Ma et al., 2023a; Pan et al., 2012). In addition, we found two observation sites in Jilin and Liaoning provinces in Northeast China from the NNDMN. At the Jilin site (124.83°E, 43.53°N), in winter months, the simulated monthly dry deposition of nitrate (atmospheric gaseous and particulate nitrate, the same as follows) was $0.07 \pm 0.10$ kg N ha$^{-1}$ month$^{-1}$, and wet deposition was $0.16 \pm 0.28$ kg N ha$^{-1}$ month$^{-1}$, with in the ranges of the observed values of $0.13 \pm 0.03$ kg N ha$^{-1}$ month$^{-1}$ for dry deposition and $0.28 \pm 0.11$ kg N ha$^{-1}$ month$^{-1}$ for wet deposition. At the Liaoning site (121.58°E, 38.92°N), the simulated dry deposition was $0.18 \pm 0.17$ kg N ha$^{-1}$ month$^{-1}$, and wet deposition was $0.79 \pm 0.32$ kg N ha$^{-1}$ month$^{-1}$, while the observed dry deposition was $0.38 \pm 0.18$ kg N ha$^{-1}$ month$^{-1}$, while wet deposition was $0.35 \pm 0.18$ kg N ha$^{-1}$ month$^{-1}$. Although at the Liaoning site, the modeled wet and dry deposition fluxes are somewhat different from the observations, their sums (i.e.,

the total deposition fluxes) are close to each other (0.97 ± 0.36 kg N ha⁻¹ month⁻¹ vs. 0.73 ± 0.25 kg N

ha⁻¹ month⁻¹) within the range of uncertainties.

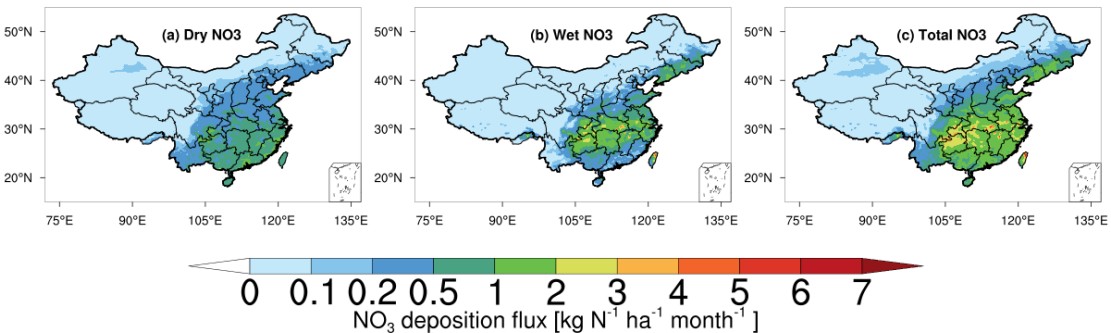

**Figure 8.** Spatial distribution of (a) dry, (b) wet, and (c) total (dry + wet) deposition fluxes (kg N ha⁻¹ month⁻¹) of oxidized nitrogen (atmospheric gaseous plus particulate nitrate) on snow simulated by WRF-
Chem in mainland China averaged over December 2017 to March 2018.

**3.4.2 Nitrate concentrations and spatial distribution in snow**

Figure 9 displays the nitrate concentration distributions across space in the top snow layer simulated

by WRF-Chem averaged for December 2017 to March 2018, with the field campaign observations of

nitrate concentration in the top snow layer (NITS). Among the four months, January 2018 had the highest

number of observations, distributed across northeastern and northwestern China. The observed data for

the other three months are primarily concentrated in the northeastern region of China. During these four

months, consistent patterns of change were identified, as the model simulating the highest NITS (> 15

µg g⁻¹) in the region spanning Northeast China (125–132° E, 40–47° N), mainly encompassing the

provinces of Heilongjiang, Jilin, and Liaoning. In addition, as far away from northeast China, the NITS

decreases and drop to less than 0.06 µg g⁻¹ at the boundary of northern China. This finding aligns with

our field campaign data, revealing elevated NITS levels (1.03–33.43 µg g⁻¹) in areas characterized by

heavy industrialization and lower concentrations (0.08–0.4 µg g⁻¹) in the northern regions of China (52°

N). From a temporal perspective, there was a significant increase in the simulated NITS in northeastern

Jilin Province from December 2017 to March 2018. This difference may be attributed to the monthly

increase in simulated nitrate deposition in this region, while snowfall slightly decreased in the

northeastern area during the same period.

The road trip during the campaign began in Inner Mongolia, which, compared to Northeast China,

exhibits relatively lower pollution levels, with most observed NITS values below 1 µg g⁻¹. The cleanest

snow samples, with concentrations in the tens of nanograms per gram range, were collected close to

China's northern border, while polluted snow was obtained from the industrialized zones of Northeast China. The WRF-Chem simulation effectively reproduces the observed notable escalation in NITS toward more polluted sites, from 0.08–0.4 μg g$^{-1}$ at 51° N to more than 10 μg g$^{-1}$ at 43° N. Both temporally and spatially, the simulation results generally align with the observations, albeit with some negative biases in relatively clean areas (e.g., Inner Mongolia).

The WRF-Chem model-simulated maximum values ranged from 7.11 to 16.58 μg g$^{-1}$, while the range of the simulated minimum values was between 0.06 and 0.21 μg g$^{-1}$. The observed maximum values varied between 9.35 and 33.43 μg g$^{-1}$, with observed minimum values falling within the range of 0.09 to 0.51 μg g$^{-1}$. In addition to the results described above, we also calculated the overall average values for the four months. The simulation results indicate an average concentration of 2.72 ± 1.34 μg g$^{-1}$, whereas the observed four-month average concentration is 3.74 ± 5.42 μg g$^{-1}$. This conclusion aligns well with the results presented with the findings of Xue et al. (2020), who also conducted observations on snowfall in northeastern China from December 2017 to March 2018. Covering the same period and region as ours, their results revealed maximum, minimum, and average nitrate concentrations in snow of 12.25, 0.08, and 3.34 ± 1.00 mg/L, respectively. Although our simulation shows a certain degree of underestimation at some sites compared to the observational results, the simulated results generally capture both the spatial patterns and magnitudes seen in the data. Regarding this underestimation, as illustrated in Figure 9, we note that there is a low bias for the NITS in high-pollution areas between December 2017 and January 2018. In particular, in high-pollution regions like Jilin Province, the model exhibited a negative bias, with an average observation-to-simulation ratio of 1.7, corresponding to a Normalized Mean Bias (NMB) of 40.29%.

In addition, we extracted the simulated values corresponding to the observations at each station and plotted them as a scatter plot (Fig. 11c). The results show that the model generally underestimates the NITS. Typically, such an underestimation of NITS could result from either underestimating the amount of snow or underestimating the flux of nitrate deposition within the snow. However, based on the snow depth simulation results, the snow amount simulation performs better, so snowfall is unlikely to be the main cause of this bias. The most likely reason for this underestimation may be that the modeled atmospheric nitrate concentration is lower than the actual concentration. Consequently, even with the same snowfall amounts, the nitrate deposition would be underestimated. To demonstrate this, we analyzed the observed atmospheric nitrate concentrations from Tracking Air Pollution in China  (Geng

et al., 2017; Liu et al., 2022b) and compared them with the simulated results. We found that in northern China, where our study area is located, the simulated atmospheric particulate nitrate concentrations were indeed lower than the observed values (Fig. S3). The low simulated nitrate concentrations in northern China may be due to incomplete atmospheric nitrate chemistry in the model. However, in other regions of southern China, such as Anhui (29.45° N - 34.55° N, 114.95° E - 119.55° E) and Fujian (23.65° N - 28.25° N, 115.95° E - 120.45° E), the simulated atmospheric nitrate concentrations closely matched the observations (Fig. S4). Thus, the effect of incomplete atmospheric nitrate chemistry in the model can be excluded in this case. Another possible reason for the low simulated nitrate concentrations in northern China could be the underestimation of $NO_x$ emissions in this region. We also compared the observed and modeled atmospheric $NO_2$ concentrations in this region and found that the model indeed underestimated the $NO_2$ concentrations (see Fig. S5). In conclusion, the underestimation of NITS in the model is most likely due to the underestimation of atmospheric nitrate concentrations, which probably originates from the model's underestimate of $NO_x$ emissions in this region.

In addition to analyzing the top snow layer, we further evaluated the model's performance by comparing the vertical distribution of nitrate in snowpack (Figure 10). Here we selected eight specific sites, and details regarding their locations and sampling information are provided in Table S1. These sites were selected because the depth intervals of observed samples in these sites are closers to the model's depth intervals. In the figure, the depth position of each point represents the midpoint of the observed or simulated depth layers, with simulations represented by stars and observations by circles. As shown in Figure 10, except sites 4 and 7, the model in general captures well the depth variations. At sites 4 and 7, the observed nitrate concentrations were much higher than other sites, and the model underestimates the observations. This pattern is similar to the model-observation comparisons of surface snow nitrate, which also indicates the model tends to underestimate surface snow nitrate at sites with high observed concentrations.

Given the substantial fluctuations in the temporal patterns of annual snowpack accumulation and the challenges in accurately predicting the occurrence of weather phenomena, aerosol releases, and deposition processes, it is judicious to compare data by utilizing the long-standing averages obtained from both actual and modeled NITS datasets across an extended timespan. Additionally, further comparisons were conducted by comparing the averaged model results within the same day with the values observed at each site on the same day. However, these analyses showed no significant alterations

(data not presented). The significant temporal fluctuations in NITS may also pose challenges when comparing monthly average values from model simulation result with field observations at particular times, a widely used method across global atmospheric modeling research (Huang et al., 2011; Qian et al., 2014; Zhao et al., 2014). The sample sites within industrial source regions are subject to increasing relative biases, with the model typically underestimating the NITS at these locations. In addition to the uncertainty in the snow accumulation process mentioned above, this difference may also be related to the challenge the model faces in capturing fine-scale variability within grid cells, which tends to be more pronounced in regions with high emissions compared to relatively clean areas.

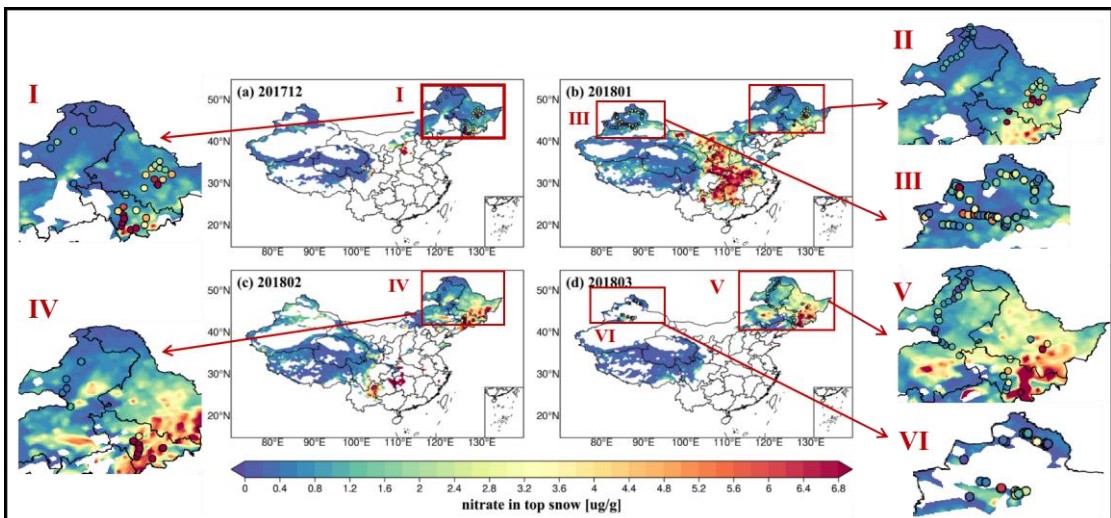

**Figure 9.** Spatial distribution of nitrate concentration in the top snow layer observed and simulated by WRF-Chem across China from December 2017 to March 2018. [Note: The background color in each figure represents the monthly average of the simulation results, while all the observations for each month are embedded in each panel.]

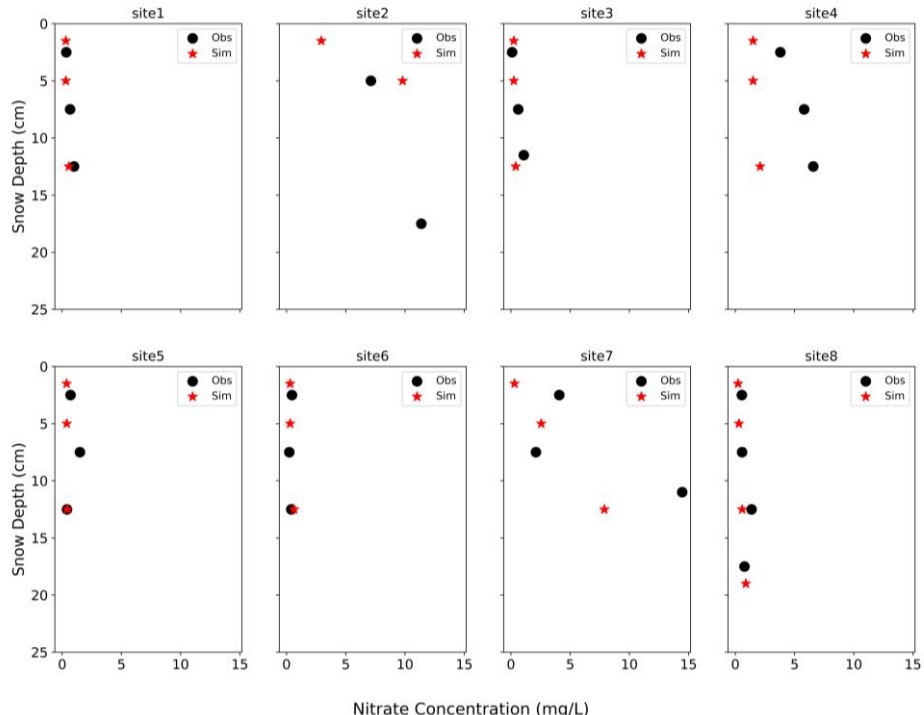

**Figure 10.** Depth profiles of the observed and simulated snow nitrate concentrations (circles for observations, stars for simulations).

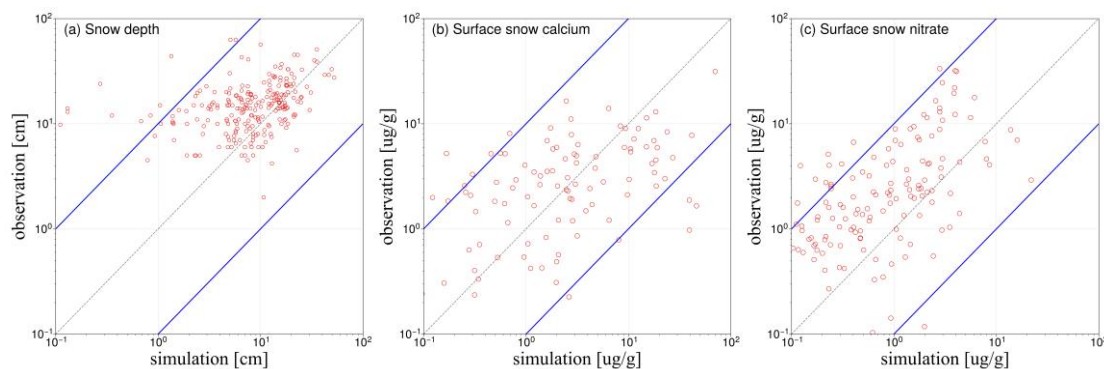

**Figure 11.** Scatter plots of the observations of **(a)** snow depth (cm), **(b)** surface snow calcium ion concentrations(ug/g)and **(c)** surface snow nitrate concentrations (ug/g) versus the corresponding WRF-Chem simulations in winter 2017–2018.

## 4 Conclusion

In this study, the WRF-Chem model was used to simulate snow cover, snow depth, and snow impurities including BC, dust and nitrate concentration in winter 2017-2018 across China. Field observations covering the same regions and periods were used to evaluate the ability of the model. In general, the model well captures the observed magnitude and spatial variations of surface temperature, snow cover, snow properties, and aerosol contents in snow. In particular, we thoroughly evaluated all simulation results with observations. Firstly, the model accurately represented the spatial patterns and

magnitudes of snow cover and snow depth. Secondly, the simulation results for the light-absorbing impurities DUST were evaluated using observational data for snow calcium ions. For snow BC concentrations, while we lack direct observations during the simulation period, we assessed the model's

performance by comparing the results with observations from the same regions during different winters (Zhao et al., 2014). Thirdly, we evaluated the simulation of snow nitrate concentrations by comparing with observational data. To assess the reasons for the discrepancies between the model and observations, we further discussed the simulation of atmospheric nitrate and its deposition fluxes. Overall, the spatial patterns and concentration levels for snow nitrate were well represented. However, in high-pollution

areas such as Jilin Province, the model exhibited larger bias, with an average observation-to-simulation ratio of 1.7, corresponding to a Normalized Mean Bias (NMB) of 40.29%.

The most likely reason for the discrepancies in NITS between the model and observations is the underestimation of atmospheric nitrate concentrations, which probably originates from the model's underestimate of NOx emissions in this region. Additionally, uncertainties in the deposition processes

(Akter et al., 2023; Huang et al., 2015; Lu and Tian, 2014), including dry and wet deposition of nitrate from the atmosphere to the snowpack, could also play a role. Furthermore, post-depositional processes could further contribute to the differences between the model and observations. These processes include snowfall dynamics, snow accumulation, and gas and aerosol scavenging in the snow  (An et al., 2022; Flanner et al., 2012; Li et al., 2022; Poschlod and Daloz, 2024; Qian et al., 2014; Zhao et al., 2014), all

of which may introduce uncertainties in the simulation of NITS. Another factor contributing to these discrepancies could be the relatively coarse model resolution, as it may not sufficiently capture the heterogeneous spatial distributions of snow and nitrate concentrations, especially when fine-scale variations are significant (Berg et al., 2024; Yu, 2013). Overall, however, the model demonstrates its ability in capturing the temporal and spatial variations in snow impurity concentrations including nitrate

in Northern China.  The considerable daily and diurnal fluctuations in simulated NITS emphasize the need for caution when comparing average values derived from the model with observations, as practiced in certain global modeling analyses. (Huang et al., 2011; Qian et al., 2014; Zhao et al., 2014).

To ensure accurate representation of aerosol contents within snow requires the model to effectively simulate the life cycle of aerosols within snowpack, as highlighted in previous studies by Flanner et al.

(2012) and Qian et al. (2014). Furthermore, uncertainties in the SNICAR model parameters must be quantified and constrained through observational data. Additionally, it is crucial for the model to

precisely replicate the atmospheric aerosol life cycle, encompassing the faithful representation of atmospheric aerosol levels and the accurate treatment of deposition mechanisms. Improvements in such model parameters and mechanisms would be necessary to further improve the agreement with

observations. Moreover, other factors such as atmospheric chemistry mechanisms may also need to be improved to better represent nitrate chemistry, which will be addressed in the next phase of this study.

Given the reasonable agreements between the model and observations, we will further incorporate snow nitrate photolysis and the subsequent emissions of $NO_2$ and/or HONO to the overlying atmosphere, investigating the potential disturbs on local to regional atmospheric chemistry with focuses on aerosol

burden which is important for atmospheric and snow radiative balances in snow cover regions, and on the potential effects on air quality originating from the winter snow cover to the downwind regions in Northern China.

**Code and data availability**

The release version of WRF-Chem can be downloaded from

http://www2.mmm.ucar.edu/wrf/users/download/get_source.html. The modified version of WRF-Chem used in this study is archived on Zenodo at https://doi.org/10.5281/zenodo.10586762. All the original data and scripts used for data processing in this study can be downloaded from https://doi.org/10.5281/zenodo.10965532.

**Author contributions**

LG conceived the study, XW conducted the model experiments and analysed the results under the supervision of LG and CZ; TC, SY and JW provided the field observational data; XW drafted the manuscript under the supervision of LG; all authors contributed to the discussion and the final version of the manuscript.

**Competing interests**

The authors declare that they have no conflict of interest.

**Acknowledgements**

L.G. and C.Z. acknowledge financial support from the National Key Research and Development Program of China (2022YFC3700701); L.G. acknowledge support from the Strategic Priority Research Program of the Chinese Academy of Sciences (XDB 41000000). T.C. acknowledge

support from the National Nature Science Foundation of China (42125604). This research was also supported by the advanced computing resources provided by the Supercomputing Center of the USTC.

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
