# Peer review of "WRF-Chem Simulations of Snow nitrate and other Physicochemical Properties in Northern China"

_Geoscientific Model Development, 2024_

## Author Comment (AC1)

- *This paper describes the development of a treatment for nitrate deposition on snow in a mesoscale model and compares the simulations with available meteorological and aerosol deposition data.   The new model developments are incremental, extending previous treatments of black carbon and other species to now include nitrate. The observations provide a unique dataset covering a wide geographic region needed to better evaluate model performance. The ability to simulate nitrate deposition is the first step to permit two-way interactions between the atmosphere and the snow. The feedback of snow nitrate on atmosphere is not covered in this paper.   This paper is clearly written and is suitable for GMD, but my preference is that more material is needed on the model development (my first major comment).*

**Response:** We are grateful to the reviewer for his/her time and efforts reviewing this manuscript. We have carefully read through the comments and made responses as well as relevant revisions in the manuscript. Please find our point-by-point response (black) and the corresponding revisions (blue) as follows.

- *My first major comment is that the details of how the deposition of nitrate on snow is handled in the model in Section 2.2.1 is far too brief.   The authors seem to point back to previous studies to indicate how other aerosol species were handled.   However, when I looked at those papers, the description of the model development there is also inadequate.   There is some description in this paper at how deposition is treated at the surface, but what is lacking is how nitrate becomes bound to snow in the atmosphere.   WRF-Chem does handle cloud-borne aerosol species, but it does not include ice-borne aerosols as indicated by the paper (line 127).   So, treating in-cloud scavenging must be parameterized in some way. Then what happens when the snow melts to rain?   Some nitrate may simply partition back to the atmosphere before it falls as snow. How is below cloud scavenging by falling snow handled in the model?   These are questions that need to be addressed.   Given the lack of details it would be difficult to reproduce the results and findings from this study using the information from this paper in its current form.*

**Response:** We thank the reviewer for this suggestion. We should have described the parameterizations with more details. To address this, in the revised manuscript, we have revised Section 2.2.1 to include additional details on the parameterization of in-cloud and below-cloud scavenging. However, it is important to note that our current model does not account for potential changes in nitrate partitioning when snow melts into rain. In this model, we assume that once nitrate enters snow or rain, it is deposited onto the surface and does not return to the atmosphere. We recognize this as a potential limitation, which could introduce some bias. While this limitation may introduce some degree of uncertainty, it is unlikely to significantly impact the modeled nitrate deposition flux. The updated section now provides more specific information on how wet deposition is calculated, including both the in-cloud and below-cloud scavenging processes. Specifically, we have revised the manuscript as following:

**From Page 6, Line 161 in the original manuscripts:** "Currently, SNICAR does not include calculations of nitrate concentrations embedded in snow. In principle, concentrations of nitrate

within each snow layer are mainly influenced by atmospheric nitrate deposition flux and snow accumulations. After deposition, layer combinations and divisions, and, in a rare case, meltwater flushing may also take effect. To quantify nitrates in snow, in this study, we parameterized nitrate concentrations in snow by considering the deposition processes of nitrate, including both dry and wet deposition. Dry deposition processes (sedimentation and turbulent mix-out) directly contribute to the accumulation of particulate and gaseous nitrate in surface snow. For gaseous nitrate (HNO₃), the dry deposition flux (kg m⁻² s⁻¹) is calculated using the following equation:

$$F_{dry\_gas} = V_{HNO3} \times C_{HNO3} \times D_{air} \tag{1}$$

where $V_{HNO3}$ is the dry deposition velocity of gaseous nitrate (m s⁻¹), $C_{HNO3}$ is the concentration of gaseous nitrate in the first (i.e., surface) layer of the atmosphere (ppmv), and $D_{air}$ is the air density in the surface layer (kg m⁻³). For particulate nitrate, the dry deposition flux is calculated for each aerosol size bin as follows:

$$F_{dry\_aer} = \sum_i (V_{no3a\_i} \times C_{no3a\_i} \times D_{air}) \tag{2}$$

where $V_{no3a\_i}$ is the dry deposition velocity of particulate nitrate in each size bin (m s⁻¹), $C_{no3a\_i}$ is the concentration of particulate nitrate in each size bin (µg kg-dryair⁻¹), and $D_{air}$ is the air density in the surface layer (kg m⁻³).

For wet deposition, it includes both in-cloud and below-cloud scavenging of gaseous nitrate and particulate nitrate including cloud-borne nitrate. In-cloud scavenging refers to the incorporation of aerosols and gases into cloud droplets as they form within clouds. Below-cloud scavenging (washout) refers to the removal of particulate and gaseous nitrate by falling hydrometeors as they descend below the cloud, where nitrate compounds are captured through mechanisms like Brownian motion, electrostatic forces, collision, and impaction, ultimately leading to their deposition on the snow surface. In this study, we estimate the amount of nitrate wet deposition by calculating the concentration changes of atmospheric total nitrate during in-cloud and below-cloud scavenging processes. For in-cloud scavenging, the concentration of cloud-borne nitrate and gaseous nitrate removed is based on the following equation:

$$\Delta C_{in-cloud} = C_{no3-cw} \times Scale_{in,\ cw} + C_{HNO3} \times Scale_{in,gas} \tag{3}$$

where $C_{no3-cw}$ is the concentration of cloud-borne nitrate aerosols in cloud, $C_{HNO3}$ is the concentration of gaseous HNO₃ in cloud, and $Scale_{in,cw}$ and $Scale_{in,gas}$ represent the scaling factors for in-cloud scavenging that indicate the amount of nitrate removed in cloud, respectively.

For below-cloud scavenging, the removal of nitrate aerosols and gases is represented as:

$$\Delta C_{below-cloud,\ i} = C_{no3a,i} \times Scale_{below,aer,i} + C_{HNO3,i} \times Scale_{below,gas,i} \tag{4}$$

where $C_{no3a,i}$ is the concentration of nitrate aerosols below the cloud in layer $i$, and $Scale_{below,aer,i}$ and $Scale_{below,gas,i}$ represent the factors for below-cloud scavenging that indicate the amount of nitrate removed by impaction-interception in each atmospheric layer. In this study, the calculations of the scavenging scales for both in-cloud and below-cloud wet removal of nitrate aerosols and gases are based on the methodologies of Easter et al. (2004) and Chapman et al. (2008). The total nitrate concentration used for wet deposition calculations is the sum of the concentrations removed during scavenging.

The wet deposition flux (kg m⁻² s⁻¹) is then calculated using the following equation:

$$F_{wet} = \Delta C_{in-cloud} + \sum_i^n \frac{(\Delta C_{below-cloud,i} \times D_{air,i} \times H_{air,\ i})}{dt} \tag{5}$$

where $D_{air,i}$ is the air density in each atmospheric layer. $H_{air,i}$ is the thickness of the atmospheric layers (m), and $dt$ is the model time step (s). The variable $n$ represents the number of atmospheric layers below the cloud.

- ***My second major comment is that it would be useful to also include an evaluation of how well the model did with regard to nitrate as a function of snow depth. The paper focuses only at the topmost snow layer. If there is a way to include a comparison of vertical distribution within the snow that would be very valuable. This will ultimately impact subsequent studies that simulate the feedback of nitrate back to the atmosphere.***

**Response:** Thank you for your suggestion regarding the evaluation of nitrate as a function of snow depth. We agree that an assessment of the vertical distribution of nitrate within the snowpack would be valuable. However, due to the mismatch between the depth stratification of our observational data and the existing depth layers in the model's snow module, it is challenging to directly compare the nitrate concentrations at specific depths. In our model, the snow layers are based on the SNICAR system, which references the thermal layers of the land surface model CLM used for thermodynamic calculations. Typically, the surface layer in the model spans 0-3 cm, the second layer 3-7 cm, and the third layer 7-18 cm. In contrast, the observational data are generally sampled in 5 cm intervals. Nevertheless, here we selected eight sites where the number of observed layers is relatively better matched with the model layers and conducted an analysis of nitrate concentration profiles with depth. A new figure (i.e., Fig. 10) is plotted to show the comparison. We also included a relevant discussion as follows in Section 3.4.2:

**From Page 22, Line 510 in the original manuscript:** "In addition to analyzing the top snow layer, we further evaluated the model's performance by comparing the vertical distribution of nitrate in snowpack (Figure 10). Here we selected eight specific sites, and details regarding their locations and sampling information are provided in Table S1. These sites were selected because the depth intervals of observed samples in these sites are closers to the model's depth intervals. In the figure, the depth position of each point represents the midpoint of the observed or simulated depth layers, with simulations represented by stars and observations by circles. As shown in Figure 10, except sites 4 and 7, the model in general captures well the depth variations. At sites 4 and 7, the observed nitrate concentrations were much higher than other sites, and the model underestimates the observations. This pattern is similar to the model-observation comparisons of surface snow nitrate, which also indicates the model tends to underestimate surface snow nitrate at sites with high observed concentrations.

Given the substantial fluctuations in the temporal patterns of annual snowpack accumulation and the challenges in accurately predicting the occurrence of weather phenomena, aerosol releases, and deposition processes, it is judicious to compare data by utilizing the long-standing averages obtained from both actual and modeled NITS datasets across an extended timespan…"

[Figure]

**Figure 10.** Depth profiles of the observed and simulated snow nitrate concentrations (circles for observations, stars for simulations).

**Table S1.** Geographical location and stratified snow depth information (observed and simulated) for the eight sites in Figure 10.

| Site | Date | Latitude | Longitude | Observed Depth (cm) | Simulated Depth (cm) |
|------|------|----------|-----------|---------------------|----------------------|
| site1 | 2018-03-10 | 51.1303 | 121.2694 | 0-5 | 0-3 |
| | | | | 5-10 | 3-7 |
| | | | | 10-15 | 7-18 |
| site2 | 2018-03-05 | 42.4668 | 127.8730 | 0-10 | 0-3 |
| | | | | 10-25 | 3-7 |
| site3 | 2018-03-10 | 51.4380 | 121.5345 | 0-5 | 0-3 |
| | | | | 5-10 | 3-7 |
| | | | | 10-13 | 7-18 |
| site4 | 2017-12-28 | 42.4668 | 127.8730 | 0-5 | 0-3 |
| | | | | 5-10 | 3-7 |
| | | | | 10-15 | 7-18 |
| site5 | 2018-03-07 | 46.6736 | 120.0428 | 0-5 | 0-3 |
| | | | | 5-10 | 3-7 |
| | | | | 10-15 | 7-18 |
| site6 | 2018-03-10 | 51.6836 | 121.8797 | 0-5 | 0-3 |
| | | | | 5-10 | 3-7 |
| | | | | 10-15 | 7-18 |
| site7 | 2018-03-13 | 46.5359 | 129.5007 | 0-5 | 0-3 |
| | | | | 5-10 | 3-7 |
| | | | | 10-12 | 7-18 |
| site8 | 2018-03-08 | 47.5509 | 119.3758 | 0-5 | 0-3 |
| | | | | 5-10 | 3-7 |
| | | | | 10-15 | 7-18 |
| | | | | 15-20 | 18-20 |

● *My third major comment is regarding the Figures, which I mention in my specific comments. The authors choose to use only spatial comparisons with the observations. Given the uncertainties in color scales, it would be more meaningful to include scatter plots or some other type of plot to provide a better quantitative assessment of the model performance. While this might add length to the paper, it is worth it. And this material could be included in supplemental information material.*

**Response:** Thanks for your constructive suggestion. We have now added or revised the figures to have scatter plots according to the following specific comments. Details are provided in responses to specific comments below.

As suggested, we made the scatter plots of modeled snow depth, surface $Ca^{2+}$ and $NO_3^-$ concentrations vs. the obversions, and added a new figure as Figure 11. We also added relevant discussions which are detailed in the responses below to each specific comments.

[Figure]

**Figure 11.** Scatter plots of the observations of **(a)** snow depth (cm), **(b)** surface snow calcium ion concentrations (ug/g) and **(c)** surface snow nitrate concentrations (ug/g) versus the corresponding WRF-Chem simulations in winter 2017–2018.

*Specific Comments:*
● *Line 78: There is an extra parenthesis.*
**Response:** Thanks for your careful check. The extra parenthesis has now been removed.

● *Line 110: change "of China" to "(USTC) of China". Suggest changing the first part of the "Distinguished …" sentence. The phrase implies the USTC version is not available to the public. I assume the authors mean the USTC developments are simply not on in the version distributed by NCAR.*
**Response:** Thanks for your suggestion. We have made suggested changes.

● *Line 111: "boasts" is a strong word. Suggest changing to "the USTC version includes supplemental functionality such as online …"*
**Response:** Thanks for your suggestion. Now we change the sentence as suggested "Unlike the version distributed by NCAR to the public, the USTC version includes supplemental functionality, such as the online diagnosis of aerosol-specific radiative forcing and the …"

● *Line 115: This version of MOSAIC is rather old and does not include newer versions that have treatments of SOA, which is often a large fraction of total particulate mass.*

*While this assumption will underestimate total PM deposition on snow, perhaps it does not matter for this time of year and when focusing on nitrate, BC, and dust observations. Some discussion on this topic seems warranted. Another area to expand upon is how calcium is treated in the model, since the authors compare predicted calcium with observations. My understanding is that by default, the model assumes a certain percentage of other inorganics are calcium, so one could tweak that ratio to better agree with observations. I am not saying that has happened here, but the reader needs to understand some details of the model to understand why the model is predicting certain species.*

**Response:** Thank you for this suggestion. 1) We acknowledge that the version of MOSAIC used in this study does not include the treatment of secondary organic aerosols (SOA), which can contribute a significant fraction of total particulate mass under certain conditions. However, during the winter months, especially in cold regions like those examined in this study, the formation and presence of SOA are typically much lower due to reduced precursor emissions, which limits SOA production. While SOA does not directly influence nitrate formation, it can affect aerosol properties, which may affect the production and phase-partition of atmospheric of nitrate. As such, we now added a brief discussion in Section 2.1 as "We note the MOSAIC aerosol scheme used in this study does not include secondary organic aerosols (SOA), which may affect the production and phase-partition of particulate nitrate"

2) Regarding the calculation of calcium ions in the model, we use a dust emission scheme where Ca and $CO_3$ emissions are scaled to the dust emission flux. In the default version of the model, the mass fractions of Ca and $CO_3$ in dust are set at 0.4% and 0.6%, respectively. However, as our research area is northern China, particularly the northwest and Loess Plateau, which is a major source area for dust, these default proportions are not representative of the real values as suggested by observations conducted in these regions which indicates the mass fraction of calcium in dust ranges from 7% to 12% in northern China (Zhang et al., 2003). In the original submission, we just used the default mass fraction in WRF-chem. In the revised manuscript, when comparing with observations, we used the average observed fraction of 9.5%. We now add a brief explanation in Section 3.3.1 as following:

**From Page 16, Line 382 in the original manuscripts:** "The modeled calcium ion content in the snow was calculated based on the proportion of calcium carbonate in the GOCART dust emission mechanism used in WRF-Chem, where calcium is assumed to constitute 0.4% of the total dust mass, and carbonate ($CO_3^{2-}$) accounts for 0.6% (Ginoux et al., 2001; Kok et al., 2014a; Kok et al., 2014b). However, as our research area is northern China, particularly the northwest and Loess Plateau, which is a major source area for dust, these default proportions are not representative of the real values as suggested by observations conducted in these regions which indicates the mass fraction of calcium in dust ranges from 7% to 12% in northern China (Zhang et al., 2003). Therefore, we used the average observed fraction of 9.5% to calculate the modeled calcium concentrations in this study. Field-observed calcium ion concentrations in the top layer of snow (CAS)…."

● *Figure 6: the color scale needs revision to have more gradients to better understand differences between the observations and simulated values. Also a scatter plot of obs vs model would be useful. As in Figure 5, it is best to zoom in a focus on the areas with observations.*

**Response:** Thanks for your suggestion. We have revised Fig. 6 to separate Dust and calcium ion concentrations in the top snow layer. Dust is now displayed in Fig. 5, while Fig. 6 focuses solely on calcium ion concentrations. Additionally, Fig. 6 has been updated to incorporate more color gradients, and, as suggested, the figure has been zoomed in to focus on the areas with observations. Also, we have added Fig. 11b, a scatter plot of observation vs. model, which is included in the revised manuscript. It is now clarified at the end of Section 3.3.1 in the revised manuscript as follows: "In addition, we extracted the simulated values corresponding to the observations at each station and plotted them in a scatter plot (Fig.11b). From the results, the simulated snow calcium ion concentrations generally fall within the same order of magnitude as the observations."

[Figure]

**Figure 5**. Spatial distribution of dust concentrations (mg/g) in the top snow layer simulated by WRF-Chem across China from December 2017 to March 2018 (a–d).

[Figure]

**Figure 6**. Spatial distribution of calcium ion concentrations (ug/g) in the top snow layer simulated by WRF-Chem, with field campaign observations embedded at specific locations for each month (e–h) across China from December 2017 to March 2018.

- *Lines 127-128: Is this process included by Chapman et al. (2008), or has it been added later?*

**Response:** Yes, this process was included by Chapman et al. (2008).

- *Line 128: by "convection" I assume you mean parameterized shallow and deep convective clouds that are subgrid scale? Please be more specific.*

**Response:** Sorry for the confusion. In this context, "convection" refers to convective transport, and we have clarified this in the revised manuscript as" The removal of aerosols by convection transport and their wet deposition via cumulus clouds are modeled according to the methods described by Zhao et al. (2013)."

- *Lines 155-159: The way the text is phrased is sounds like snow photolysis has been included, but elsewhere the authors indicate that the feedback to the atmosphere is not included. Suggest revising this text to eliminate the possible confusion introduced.*

**Response:** We are sorry for making this text confusing. Now we revise the statement to more clearly express our point as:"Physical properties are used to simulate radiative transfer in snow. While nitrate and other impurities in snow also influence radiative transfer snow, and especially nitrate in snow is the source of snow-sourced $NO_x$. Currently, all other components (e.g., BC, dust) but not nitrate have been included in SNICAR and parameterized by Zhao et al. (2014). In this study, we parameterized and included snow nitrate concentration in simulation."

- *Figure 1: Samples is misspelled in the figure. Put a space between the year and month. What is the difference between red dots and red stars?*

**Response:** Thanks for your careful check. Figure 1 is now revised as suggested. The red dots are actually yellow dots overlapped by red stars, so that they appear as red dots. To avoid this, in the revised manuscript, we enlarged the yellow dots and reduced the size of the stars, and changed the solid circles to hollow.

[Figure]

**Figure 1.** Sampling points along the road trip from December 2017 to March 2018 are marked with different colors to represent different months. Color indicates different months of the observations.

- *Line 288: This sentence is provided as a motivation for the model evaluation of meteorological variables. But the accuracy is a bit more complicated than the authors note here. The accuracy will also depend on how well the model simulates synoptic circulations and the treatment of cloud microphysics, in addition to other factors.*

**Response:** Thanks for your insightful comment. We agree that the accuracy of snow simulation is influenced by more than just temperature and precipitation, including factors such as synoptic circulations and cloud microphysics. Indeed, this sentence was intended as a motivation for evaluating meteorological variables, and we acknowledge that our initial expression may have been imprecise. We have now revised the sentence in the manuscript as: "Two key factors affecting the snow simulation are surface temperature and snow precipitation."

- *Lines 290-291: I thought the Morrison microphysics scheme contains a snow specie, but there the authors suggest otherwise. Perhaps a bit more discussion is needed here.*

**Response:** Thanks for pointing it out. We are sorry for the confusion. You are correct that the Morrison microphysics scheme in WRF-Chem does simulate snow, but it does not simulate snowfall separately at the surface layer, where all precipitation is treated as rainfall but at temperatures below $0\,^{\circ}\mathrm{C}$ at the surface rainfall is then treated as snowfall. We have now removed the original sentence and provided a more specific explanation in the following response in the manuscript.

- *Figures 2-3: Perhaps it would be useful to include a scatter plot of obs vs model as an extra panel. The dots could be colored by region to show performance differences between northern and southern China.*

**Response:** Thanks for this. We have added a scatter plot as an extra panel in Figure 2 of the revised manuscript, with dots colored by region to illustrate performance differences between northern and southern China. Regarding Figure 3, we have conducted a further review and discovered that the observed precipitation data represents only liquid precipitation, without accounting for solid precipitation (i.e., snow). Currently, we can't find available observation of surface snow fall data for model evaluation. Based on this consideration, we think that evaluating liquid precipitation has limited relevance to snow simulation, therefore, we removed this figure in the revised manuscript. Now we have included the explanation and added much analyses in Section 3.1 in the revised manuscript:

**Page 11, Line 288 in the original manuscripts:** "Two key factors affecting the snow simulation are surface temperature and snow precipitation. Since there are no publicly available observation data for surface snow precipitation, only temperature was compared with observations for model evaluation.

Figure 2 displays the 2 m temperature patterns across China simulated by WRF-Chem and observed, with the left panel showing the spatial distribution of temperature, and the right panel illustrating the scatter plot comparison between simulation and observation. The background color in the left panel represents the average simulated values from December 2017 to March 2018, while the right panel shows the daily averages from the simulation corresponding to the observation dates at each station. The scatter plot also distinguishes between regions, with orange dots representing northern China and blue dots representing southern China. Daily 2 m temperature data from December 2017 to March 2018 at 415 sites in China were sourced from the National Oceanic and Atmospheric Administration (NOAA). Based on the graph, it is evident that the model accurately depicts the spatial patterns and fluctuations in the 2 m temperature, aligning well with the observed data. Furthermore, the simulation accurately represents the notable decrease in the 2 m temperature as latitude increases, ranging from near freezing levels to approximately -30°C. From the scatter plot on the right, it can be observed that the model generally performs well in simulating the 2 m temperature, closely aligning with the observed data. However, there is a slight underestimation of temperature for southern China and a slight overestimation for northern China. Such systematic biases have also been reported in other studies (Gao, 2020; Gao et al., 2022; Kong et al., 2019; Yu et al., 2011). These discrepancies may be attributed to the complexity of regional climate factors, such as varying land surface characteristics, boundary layer processes, and the challenges of accurately simulating localized weather phenomena like cold fronts or temperature inversions in certain regions. Furthermore, differences in the representation of terrain and vegetation between the model and reality could contribute to these systematic errors, particularly in regions with complex topography (Gutowski et al., 2020)."

[Figure]

**Figure 2**. Spatial distribution and correlation of 2 m temperature observed and simulated by WRF-Chem across China from December 2017 to March 2018. [Left: The background color represents the average simulated value. Right: Simulated values are the daily averages corresponding to the observation dates at each station.]

● *Figure 4: Fraction is misspelled in the figure.   It would be better if the MODIS and WRF panels were together (maybe left vs right), so that reader can compare them easier.   Has the MODIS data been averaged to the 36 km WRF grid? This would provide a fairer comparison.   Then a scatter plot (perhaps by region) would provide a more meaningful quantification of the difference.*

**Response:** We thank the reviewer for this suggestion. We have now averaged the MODIS data to the 36 km WRF-Chem grid and adjusted the panels to a left-right format for easier comparison. To better compare the differences between the simulation and observations, we have added a third column showing the difference map, calculated as the simulation minus the observation. Given that the MODIS resolution is 500 m, it can more effectively capture subtle variations in terrain and vegetation cover. While the overall spatial trends of the observations and simulations may appear similar, the absolute differences in individual grid boxes can be significant. This is due to the higher precision of the MODIS products, which take into account various surface types and vegetation coverage, whereas the WRF-Chem model may not capture these factors in as much detail. Given these considerations, we feel that a scatter plot may not adequately represent the differences in this context. Therefore, to provide a clearer evaluation, we have included a difference map in the third column to better illustrate the simulation-observation discrepancies. To illustrate this, we revised this paragraph to include more discussions as follows:

**From Page 13, Line 321 in the original manuscripts:** "Before analyzing the patterns of light-absorbing impurities in snow, it is crucial to assess the simulated snow cover produced by WRF-Chem. Figure 4 (now Fig. 3) shows the spatial patterns of snow cover (the percentage of land area with snow at each grid cell) from the WRF-Chem simulation (first column), MODIS-based observational data (second column), and the difference between the two (third column) from December 2017 to March 2018, providing the average results for each month. The MODIS data has been averaged to the 36 km WRF grid for a fairer comparison. The third column shows the difference map, calculated as the WRF-Chem

simulation minus the MODIS observations, highlighting areas where the model either overestimates or underestimates snow cover. Snow cover is defined as the snow fraction [0-1], which represents the percentage of land area with snow at each grid point. Both simulations and observations indicate that snow cover is concentrated primarily in China's northeastern, northwestern, and Qinghai–Tibet Plateau regions. The distribution of snow cover generally follows the temperature pattern. Areas with lower temperatures tend to have greater snow cover. The highest snow cover percentage, up to 90%, is observed in the northeastern region. Both the observations and simulations reveal snow accumulations in central China in January 2018. The difference between the simulation and MODIS data in the third column reveal systematic biases. In particular, the WRF-Chem model tends to overestimate snow cover in parts of northern China, especially in regions with complex terrain or higher altitudes. This overestimation could be attributed to the model's potential oversensitivity to cold temperatures or its overestimation of snowfall in these colder regions. Complex terrain can also challenge the model's ability to accurately simulate microclimatic conditions, leading to discrepancies in snow cover estimates. Conversely, in southern and central China, the model underestimates snow cover, likely due to limitations in how WRF-Chem handles snow accumulation and melting in warmer areas. Overall, the model appears to reasonably capture the stable snow cover in most of the regions of interest, though some discrepancies remain related to small-scale surface features caused by terrain, with most biases staying within 30%."

[Figure]

**Figure 4.** Spatial distribution of snow cover simulated by WRF-Chem and observed from MODIS-based data across China from December 2017 to March 2018. The data presented are monthly averages, and the third column shows the difference calculated as the simulation minus the observation.

- *Figure 5: This figure could be improved by only including one China-wide plot that only shows the boxes. The small panels are more important and could then be larger.*

**Response:** Thanks for this suggestion. However, since each China-wide plot represents a different month with varying observation locations, the boxes differ as well. For this reason, we think it is necessary to retain the current format of Fig. 5 (now Fig. 4). Now we have added scatter plots for snow depth in the revised manuscript as shown in **Fig. 11a**. However, during our analysis of snow depth, we identified a systematic underestimation in the simulation results. Upon further investigation, we found that the snow density value used in the model to calculate snow depth **was a constant value of 250 kg/m³**, which is higher than the observed snow density in northern China, **typically around 180 kg/m³**. We think that this discrepancy in snow density may be a contributing factor to the underestimation of snow

depth in the model. To address this, we adjusted the simulated snow depth using the observed snow density values. We have updated Figure 5 (now Fig. 4) and the corresponding scatter plot results in the revised manuscript. It is now clarified at the end of Section 3.3.2 in the revised manuscript as follows: "In addition, we extracted the simulated values corresponding to the observations at each station, as shown in Fig. 10a. From the results, most of the simulated snow depths align reasonably well with the observations, though underestimation is evident in some areas, particularly in regions with lower snow depths."

[Figure]

**Figure 5.** Spatial distribution of snow depth (cm) observed and simulated by WRF-Chem and across China from December 2017 to March 2018. [Note: The background color in each figure represents the monthly average of the simulation results, while all the observations for each month are embedded in each panel.]

- *Line 406: The statement "the model agreed well with the observations" is too vague and subjective.   Please give some #'s or better description.*

**Response:** Thank you for raising this issue. Now we add more description of this statement in Section 3.3.2 as follows:

**From Page 18, Line 403 in the original manuscripts:** "We do not have BC observations during the simulation period, but there were observations in other winters for the same regions. We compared our simulations with those observations, and the results show that both the magnitude and the spatial patterns of our simulated BC concentrations are consistent with the observed values reported in the literature (Zhao et al., 2014). In addition, Zhao et al. (2014) also used the same model and framework to simulate BC concentrations during the observational period. Their study showed reasonable agreement with a median model-to-observation ratio of 1.03. In the vicinity of approximately 40° N and 125° E in Northeast China, as depicted in Fig.7, the highest concentrations of BC in the top snow layer…."

- *Lines 401-420:   Since there are no BC obs, what is the value of this section to the main goal of the paper to describe nitrate?   This section could be deleted.   If not, provide a better justification for including it.*

**Response:** BC is an important light-absorbing impurity. In order to process the model nitrate photolysis, we need to know its concentration. Therefore, in Section 3.3, we aim to evaluate the model's performance in simulating snow BC contents. We don't have BC observations during the simulation period, but there were observations in other winters for the same regions. We compared our simulations with those observations, and the results show that both the magnitude and the spatial patterns of our simulated BC concentrations are consistent with the observed values reported in the literature (Zhao et al., 2014). In addition, Zhao et al. (2014) also used the same model and framework to simulate BC concentrations during the observational period. Their study showed reasonable agreement with a median model-to-observation ratio of 1.03. Our simulated results are consistent with those of Zhao et al. (2014) in terms of the order of magnitude and the spatial distribution. This comparison suggests that the model's performance in simulating snow BC is reasonably close to the observed values, indicating that the model operates within an acceptable range for snow BC simulation. Therefore, we believe it is necessary to retain the evaluation of BC in Section 3.3.2.

- *Figure 8: There is something in the lower right corner that is unreadable. If unimportant it should be removed.*

**Response:** Thank you for pointing out this. However, I'm not sure if you are referring to the South China Sea inset in the lower right corner which highlighted the area with a red arrow. If so, this inset makes the map of China complete and we hope to keep it.

[Figure]

**Figure 8.** Spatial distribution of (a) dry, (b) wet, and (c) total (dry + wet) deposition fluxes (kg N ha$^{-1}$ month$^{-1}$) of oxidized nitrogen (atmospheric gaseous plus particulate nitrate) on snow simulated by WRF-Chem in mainland China averaged over December 2017 to March 2018.

- *Figure 9: Same comment as Figure 5. Again a scatter plot comparing obs and model is needed. Another figure is needed to show trends from month to month. It is hard to infer this, but the authors are claiming good predictions in the trends in the conclusions.*

**Response:** We thank the reviewer for providing this suggestion. Because of this suggestion, we reexamined Figure 9, and found there was a typo in old Fig. 9, so we corrected it and the new Fig. 9 is shown below. Regarding the suggestion to have one pot in the center, we didn't make change because each China-wide plot represents a different month with varying observation locations, the boxes differ as well. For this reason, we believe it is necessary to retain the current format of Fig. 9. Regarding your comment on showing trends from month to

month, we would like to clarify that the term "spatial trends" in our manuscript was intended to refer to spatial variations. We apologize for any confusion caused by the choice of wording. We have now revised all instances of "**spatial trends**" to "**spatial patterns**" to more accurately reflect the intended meaning in the revised manuscript.

[Figure]

**Old Figure 9.**

[Figure]

**New Figure 9.** Spatial distribution of nitrate concentration in the top snow layer observed and simulated by WRF-Chem across China from December 2017 to March 2018. [Note: The background color in each figure represents the monthly average of the simulation results, while all the observations for each month are embedded in each panel.]

In addition, also as suggested, we have added scatter plots for snow surface nitrate in the revised manuscript as shown in Fig. 11c.

However, from Fig. 11c, we notice that the model generally underestimates the surface snow nitrate concentrations. Therefore, we further discussed the possible reasons for this underestimation. We have added this discussion to Section 3.4.2 in the revised manuscript as following:

**From Page 22, Line 504 in the original manuscript:**

"Regarding this underestimation, as illustrated in Figure 9, we note that there is a low bias for the NITS in high-pollution areas between December 2017 and January 2018. In particular, in high-pollution regions like Jilin Province, the model exhibited a negative bias, with an average observation-to-simulation ratio of 1.7, corresponding to a Normalized Mean Bias (NMB) of 40.29%.

In addition, we extracted the simulated values corresponding to the observations at each station and plotted them as a scatter plot (Fig. 11c). The results show that the model generally underestimates the NITS. Typically, such an underestimation of NITS could result from either underestimating the amount of snow or underestimating the flux of nitrate deposition within the snow. However, based on the snow depth simulation results, the snow amount simulation performs better, so snowfall is unlikely to be the main cause of this bias. The most likely reason for this underestimation may be that the modeled atmospheric nitrate concentration is lower than the actual concentration. Consequently, even with the same snowfall amounts, the nitrate deposition would be underestimated. To demonstrate this, we analyzed the observed atmospheric nitrate concentrations from Tracking Air Pollution in China (Geng et al., 2017; Liu et al., 2022) and compared them with the simulated results. We found that in northern China, where our study area is located, the simulated atmospheric particulate nitrate concentrations were indeed lower than the observed values (Fig. S3). The low simulated nitrate concentrations in northern China may be due to incomplete atmospheric nitrate chemistry in the model. However, in other regions of southern China, such as Anhui (29.45° N - 34.55° N, 114.95° E - 119.55° E) and Fujian (23.65° N - 28.25° N, 115.95° E - 120.45° E), the simulated atmospheric nitrate concentrations closely matched the observations (Fig. S4). Thus, the effect of incomplete atmospheric nitrate chemistry in the model can be excluded in this case. Another possible reason for the low simulated nitrate concentrations in northern China could be the underestimation of $NO_x$ emissions in this region. We also compared the observed and modeled atmospheric $NO_2$ concentrations in this region and found that the model indeed underestimated the $NO_2$ concentrations (see Fig. S5). In conclusion, the underestimation of NITS in the model is most likely due to the underestimation of atmospheric nitrate concentrations, which probably originates from the model's underestimate of $NO_x$ emissions in this region.

In addition to analyzing the top snow layer, we further evaluated the model's performance by comparing the vertical distribution..."

[Figure]

**Figure S3.** Observed atmospheric nitrate concentrations in **(a)** Heilongjiang and **(b)** Jilin versus the corresponding WRF-Chem simulations for January 2018 in Northern China.

**Figure S4.** Observed atmospheric nitrate concentrations in **(a)** Anhui and **(b)** Fujian versus the corresponding WRF-Chem simulations for January 2018 in Southern China.

**201801 NO₂ in Northern China**

[Figure]

**Figure S5.** Observed atmospheric NO₂ concentrations in **(a)** Heilongjiang and **(b)** Jilin versus the corresponding WRF-Chem simulations for January 2018 in Northern China.

- *Lines 534-535: This statement is subjective and would be useful to include some #'s on bias, correlation, etc. regarding what constitutes "effectively replicates"*

**Response:** We thank the reviewer for raising this. We are sorry for the unclear expression in the original sentence. To better convey our intended meaning, we have revised the sentence in the updated manuscript to: "In general, the model well captures the observed magnitude and spatial variations of surface temperature, snow cover, snow properties, and aerosol contents in snow."

- *Line 538: this statement talks about validation with BC, but there were no observed BC in this study. So, are the authors implying the BC deposition is similar to other studies in a climatological sense? This needs more clarification.*

**Response:** Sorry for the confusion. What we intended to express is that the model's simulation of BC and DUST in snow were assessed through various methods to validate the reliability of the results. For DUST, we used snow calcium ion observations for evaluation. For snow BC concentrations, although we lack observations for the simulation period, we assessed the model by comparing the results with observations from the same regions during different winters (Zhao et al., 2014). The specific evaluation and bias results can be found in our response to the above comment. Now these points are clarified in Section 4 in the revised manuscript as:

**Page 23, Line 538 in the original manuscript:** "Secondly, the simulation results for the light-absorbing impurities DUST were evaluated using observational data for snow calcium ions. For snow BC concentrations, while we lack direct observations during the simulation period, we assessed the model's performance by comparing the results with observations from the same regions during different winters (Zhao et al., 2014)."

- *Line 540: "validity of the results" is an another vague, subjective statement. What do the authors mean by "assessing the simulation outcomes." It seems to me the assessing*

*part was the evaluation of the simulated nitrate with observations, which is part of the "validity of the results" phrase. So the authors need to better communicate a message here.*

**Response:** We thank the reviewer for raising this. We are sorry for the unclear expression in the original sentence. We have now clarified this in the revised manuscript as "Thirdly, we evaluated the simulation of snow nitrate concentrations by comparing with observational data. To assess the reasons for the discrepancies between the model and observations, we further discussed the simulation of atmospheric nitrate and its deposition fluxes."

- *Line 541: What is "slight"? Can you give a percentage? That would be more meaningful.*

**Response:** We thank the reviewer for the thoughtful suggestions. Now we have added a quantitative bias to the statement in the revised manuscript as "Overall, the spatial trends and concentration levels for snow nitrate were well represented. However, in high-pollution areas such as Jilin Province, the model exhibited larger bias, with an average observation-to-simulation ratio of 1.7, corresponding to a Normalized Mean Bias (NMB) of 40.29%."

- *Line 544: Could part of this be due to not including the feedback of nitrate back to the atmosphere? If so, that would mean that the present results should be biased high compared to observations. But that doesn't seem to be the case at all locations.*

**Response:** Thank you. We agree that not including the photolysis of nitrate in snow could indeed contribute to the discrepancy between the model and observations. However, this process tends to decrease snow nitrate concentrations, while our model results show an underestimation of snow nitrate concentrations, which is opposite to the current bias in the model. Therefore, the discrepancy between the model and observations here is not due to the lack of nitrate feedback. As discussed in the earlier response, the primary reason for the underestimation is most likely due to the underestimation of atmospheric nitrate concentrations, which probably originates from the model's underestimate of $NO_x$ emissions in this region. Additionally, other factors such as atmospheric chemistry mechanisms may also need to be improved to better represent nitrate chemistry. These will be addressed in the next phase of this study.

- *Line 545: Is there a stray "w"?*

**Response:** Thank the reviewer for the detailed check. Now we have removed this additional 'w'.

**Reference**

[revised manuscript text omitted]

---

## Author Comment (AC2)

- *Snow is a key component of the cryosphere and has significant impacts on surface energy balance, hydrology, atmospheric circulation, and more. Moreover, snow is significant in atmospheric chemistry, where snow impurities such as nitrate are sensitive to sunlight and can be photolyzed to emit reactive species including NO2 and HONO, which can significantly disturb atmospheric chemistry, especially in pristine regions. An accurate description of the emission and atmospheric consequences of snow-emitted reactive species is hence important for assessing the atmosphere environment. To address this issue, the authors parameterized atmospheric nitrate deposition and its distributions in snow using WRF-Chem model, the performance of the simulations in snow depth, and BC, dust and nitrate concentrations are well validated by field observations in northern China. Overall, this paper is well written and will be very helpful to the related research communities to improve the understanding of snow-atmosphere interactions and its influence on environments. I think this work is suitable for publication in GMD if the following concerns can be addressed:*

**Response:** We are grateful to the reviewer for his/her time and efforts reviewing this manuscript. We have carefully read through the comments and made responses as well as relevant revisions in the manuscript. Please find our point-by-point response (black) and the corresponding revisions (blue) below.

*Major comments*

- *1. You point out that to simulate snow nitrate photolysis and its impacts on overlying atmospheric chemistry, one need to obtain snow cover, snow depth, and snow physical and chemical properties, including snow density; impurities, including BC, dust; and nitrate. Other studies have parameterized most factors except snow nitrate concentration, which was the primary contribution of your work. However, your title was "Simulations of Snow Physicochemical Properties in Northern China using WRF-C", and the abstract includes much descriptions about the simulation and validations of snow cover, snow depth, and BC and dust concentrations, which ware not belonging to your work. In contrast, the description about snow nitrate simulation, the primary contribution of your work, was not enough. So, I suggest some necessary revisions to the title and abstract to emphasis your highlight on snow nitrate simulation. For example, in your abstract, the quantitative performance in nitrate concentration simulation, the bias analysis, and possible bias sources should be included to show the readers how good is your simulation. In addition, the results should more focus on snow nitrate simulation.*

**Response:** We sincerely thank the reviewer for the insightful comments and suggestions. We agree with the reviewer that the title and abstract should more clearly emphasize the primary contribution of our work, specifically the snow nitrate simulation. In light of this, we have revised the title and abstract to reflect this focus.

**The new title is:**

"WRF-Chem simulations of snow nitrate and other Physicochemical Properties in Northern China

"

**The new abstract is:**

"Snow is a key component of the cryosphere and has significant impacts on surface energy balance, hydrology, atmospheric circulation, and etc. In addition, numerous studies have indicated that snow impurities, especially nitrate, are sensitive to sunlight and can be photolyzed to emit reactive species including $NO_2$ and HONO, which serve as precursors of $O_3$ and radicals and disturb the overlying atmospheric chemistry. This makes snow an important reservoir of reactive species, especially in remote and pristine regions with limited anthropogenic emissions. The magnitude of snow chemical emissions is also influenced by snow physical properties, including snow depth, density and concentrations of light-absorbing impurities (e.g., BC and dust). Exploring and elucidating the emissions and atmospheric consequences of the snow-sourced reactive species require a global or regional model with a snow module. Here, we parameterized atmospheric nitrate deposition and its distributions in snow using a regional chemical transport model, i.e., the WRF-Chem (Weather Research and Forecasting Model coupled with Chemistry) model, and evaluated its performance in simulating snow cover, snow depth, and BC, dust, and nitrate concentrations with field observations in northern China, one of the regions with dense and prolonged snow cover. In general, the model simulated spatial variability of nitrate mass concentrations in the top snow layer (hereafter NITS) are consistent with observations. Simulated NITS values in Northeast China from December 2017 to March 2018 had a maximum range of 7.11–16.58 µg g$^{-1}$, minimum range of 0.06–0.21 µg g$^{-1}$, and a four-month average of 2.72 ± 1.34 µg g$^{-1}$. In comparison, observed values showed a maximum range of 9.35–33.43 µg g$^{-1}$, minimum range of 0.09–0.51 µg g$^{-1}$, and an average of 3.74 ± 5.42 µg g$^{-1}$. The model results show an underestimation especially in regions closes to large cities in northeastern China, most likely due to the underestimation of $NO_x$ emissions in these regions. Additionally, nitrate deposition, snowpack accumulation processes, and challenges in capturing fine-scale emission variability may also contribute to the bias. The results illustrate the ability of WRF-Chem in simulating snow properties including concentrations of reservoir species in northern China, and in the future, we will incorporate snow nitrate photolysis in the model, exploring the emissions of snow $NO_x$ from nitrate photolysis and the impacts on local to regional atmospheric chemistry and air pollutant transformations."

*The descriptions on method were not clear:*
- *(a) Line 180, from Equations 2 and 3, horizontal diffusion was not concluded in wet deposition calculation, is its influence was insignificant?*

**Response:** Thanks for the comment. In WRF-Chem, horizontal diffusion of chemical species is included during transport processes. By the time wet deposition is calculated, chemical concentrations have already been influenced by horizontal diffusion, so its effects are inherently included in the deposition calculation.

- *(b) Line 192, form Equation 4, the unit of MNITS should be same to $\Delta F \times dtime/\Delta Wsno$. However, in Equation 5, the unit of MNITS was same to $\Delta F \times dtime/\Delta Wsno \times \Delta t$.*

**Response:** Thanks for your careful check. We are sorry to make this mistake. Here we multiplied

by an extra $\Delta t$. The term $\Delta t$ already represents a cumulative value for the wet and dry deposition which means that the additional multiplication by $\Delta t$ was incorrect. We have revised the formula by removing it in the Equation 5 (now Equation 7) as: "

$$M_{NITS}^{new} = M_{NITS} + \frac{\Delta F \times dtime}{\Delta W_{sno}} \tag{7}"$$

- **(c) Line 193, ΔF is the cumulative wet and dry deposition of atmospheric nitrate during the entire period between the newly fallen snow and the previous time step. This means the unit of ΔF was kg m-2. If so, the unit of the second term in Equation 5 was not kg kg-1. Please check.**

**Response:** Thank you for pointing out the discrepancy. We have corrected it.

- **(d) Line 219-220, you mentioned "the nitrate concentrations in each snow layer are determined by factors such as atmospheric deposition rates, the amount of new snowfall, layer combinations and divisions, and meltwater flushing (Oleson et al., 2010b; Flanner et al., 2012; Flanner et al., 200 ", how did you consider the layer combinations and divisions in your simulation.**

**Response:** We thank the reviewer for the comments. In our simulation, which calculates nitrate concentrations in snow, when new snowfall occurs, if a snow layer has nearly melted or its thickness falls below the minimum threshold, it is combined with the adjacent upper or lower layer to streamline the simulation. Conversely, if the snow layer exceeds the maximum thickness, it is subdivided into two layers of equal thickness, retaining the liquid water, ice content, and temperature of the original layer. This approach follows the snow layering system from SNICAR, which is based on the thermal layers used for thermodynamic calculations in the CLM land surface model (Flanner et al., 2012; Flanner and Zender, 2005; Flanner et al., 2007; Oleson et al., 2010). The model typically defines snow layer thicknesses as follows: the surface layer spans 0–3 cm, the second layer 3–7 cm, the third layer 7–18 cm, the fourth layer 18–41 cm, and the bottom layer exceeds 41 cm.

- **(e) Line 185, you assumed a mix with the top 2cm layer? Do you have any references? For dry deposition, I can agree with your assumption, but for wet deposition, such an assumption may induce significant bias.**

**Response:** Thank you for pointing this out. We apologize for the lack of clarity in our original text. We intended to express that, upon deposition, nitrate is mixed instantly and uniformly in the model surface layer, which never exceeds 3 cm in thickness. In the original manuscript, we mistakenly stated 2 cm. We have now revised the sentence in the revised manuscript:
Page 7, Line 185 in the original manuscript: "After deposition, nitrate is mixed instantly and uniformly in the model surface layer, which never exceeds 3 cm thick." This indicates that we are adding nitrate to the model's surface layer, which is defined as 0-3 cm based on the SNICAR layering approach (Flanner et al., 2012; Flanner and Zender, 2005; Flanner et al., 2007; Oleson et al., 2010).

- **(f) Line 212, the scavenging ratio for nitrate was assigned to 0.2. From my knowledge, the nitrate was much soluble. Assigning a low scavenging ratio should add more**

*discussions.*

**Response:** We agree with the reviewer that nitrate is highly hydrophilic and easily soluble in water. The value of 0.20 used in our study is derived from previous assumptions made by Flanner et al. (2012) and Zhao et al. (2014) regarding BC, which are generally reasonable compared to observations (Doherty et al., 2013). We acknowledge that this assumption may be oversimplified, as the value for nitrate is uncertain, and 0.20 may indeed underestimate the scavenging of nitrate. However, for this process to be effectively impactful, significant melting would need to occur. During our simulation period, temperatures in northern China were consistently low, primarily below $0\,°C$, and significant melting did not take place. Therefore, we believe the impact of this assumption is minimal in this context. We have now included a more detailed discussion in the revised manuscript, specifically in Section 2.2.2:

Page 8, Line 211 in the original manuscript: " $D$ represents the combined effect of total atmospheric particulate and gaseous nitrate deposition, which is specifically added to the surface layer of the snowpack. In this study, following Flanner et al. (2012) and Zhao et al. (2014), the scavenging ratio (k) for nitrate is assumed to be 0.2. This value is highly uncertain for nitrate and needs to be constrained by future observations (Flanner et al., 2012; Qian et al., 2014; Zhao et al., 2014). However, for this process to be effectively impactful, significant melting would need to occur. During our simulation period, temperatures in northern China were consistently low, primarily below 0°C, and significant melting did not take place. Therefore, we believe the impact of this assumption is minimal in this context. It is worth noting that the portion of nitrate mass lost through meltwater from the bottom layer of snow is considered to be removed from the snowpack and is not accounted for within the model."

- *(g) In Equation 6, what did $q_{i+1}c_{i+1}$ represent, please clarify.*

**Response:** Thanks for this comment. In Equation 6 (now Equation 9), $q_{i+1}c_{i+1}$ represents the mass flux of water leaving layer $i+1$ (the layer above) multiplied by the concentration of nitrate in that same layer $i+1$. This term accounts for the transfer of nitrate from the upper layer to the current layer $i$, as meltwater moves downward through the snowpack. Similarly, $q_ic_i$ represents the mass flux of water leaving layer i multiplied by the nitrate concentration in layer i. The change in nitrate mass in layer i influenced by the amount of nitrate coming from layer i+1 and the amount leaving layer $i$. We have clarified this in the revised manuscript as "The term $q_{i+1}c_{i+1}$ represents the mass flux of water leaving the layer above (i+1) multiplied by the concentration of nitrate in that layer, accounting for the transfer of nitrate from the upper layer to the current layer."

*Minor comments:*
- *Suggest to add scatter plots of simulated versus observed data for simulation validations, especially for snow nitrate.*

**Response:** Thanks for this suggestion. Another reviewer had a similar comment, and we have now added scatter plots for snow surface nitrate in the revised manuscript as shown in Fig. 11c.

[Figure]

**Figure 11.** Scatter plots of the observations of **(a)** snow depth (cm), **(b)** surface snow calcium ion concentrations (ug/g) and **(c)** surface snow nitrate concentrations (ug/g) versus the corresponding WRF-Chem simulations in winter 2017–2018.

However, from Fig. 11c, we notice that the model generally underestimates the surface snow nitrate concentrations. Therefore, we further discussed the possible reasons for this underestimation. We have added this discussion to Section 3.4.2 in the revised manuscript as following:

**From Page 22, Line 505 in the original manuscript:**

"Regarding this underestimation, as illustrated in Figure 9, we note that there is a low bias for the NITS in high-pollution areas between December 2017 and January 2018. In particular, in high-pollution regions like Jilin Province, the model exhibited a negative bias, with an average observation-to-simulation ratio of 1.7, corresponding to a Normalized Mean Bias (NMB) of 40.29%.

In addition, we extracted the simulated values corresponding to the observations at each station and plotted them as a scatter plot (Fig. 11c). The results show that the model generally underestimates the NITS. Typically, such an underestimation of NITS could result from either underestimating the amount of snow or underestimating the flux of nitrate deposition within the snow. However, based on the snow depth simulation results, the snow amount simulation performs better, so snowfall is unlikely to be the main cause of this bias. The most likely reason for this underestimation may be that the modeled atmospheric nitrate concentration is lower than the actual concentration. Consequently, even with the same snowfall amounts, the nitrate deposition would be underestimated. To demonstrate this, we analyzed the observed atmospheric nitrate concentrations from Tracking Air Pollution in China (Geng et al., 2017; Liu et al., 2022) and compared them with the simulated results. We found that in northern China, where our study area is located, the simulated atmospheric particulate nitrate concentrations were indeed lower than the observed values (Fig. S3). The low simulated nitrate concentrations in northern China may be due to incomplete atmospheric nitrate chemistry in the model. However, in other regions of southern China, such as Anhui (29.45° N - 34.55° N, 114.95° E - 119.55° E) and Fujian (23.65° N - 28.25° N, 115.95° E - 120.45° E), the simulated atmospheric nitrate concentrations closely matched the observations (Fig. S4). Thus, the effect of incomplete atmospheric nitrate chemistry in the

model can be excluded in this case. Another possible reason for the low simulated nitrate concentrations in northern China could be the underestimation of $NO_x$ emissions in this region. We also compared the observed and modeled atmospheric $NO_2$ concentrations in this region and found that the model indeed underestimated the $NO_2$ concentrations (see Fig. S5). In conclusion, the underestimation of NITS in the model is most likely due to the underestimation of atmospheric nitrate concentrations, which probably originates from the model's underestimate of $NO_x$ emissions in this region.

In addition to analyzing the top snow layer, we further evaluated the model's performance by comparing the vertical distribution..."

- *Line 415, BCS increase during the melting period should be mainly due to melt enrichment (Doherty et al., 2013)*

**Response:** Thank you for this comment. We agree that the rise in BCS should primarily be attributed to melt enrichment, as referenced by Doherty et al. (2013). We have now revised the text to clarify this point as "As the snow begins to melt, BCS continues to rise primarily due to melt enrichment, where melting snow concentrates BC near the snow surface (Doherty et al., 2013). This effect is further enhanced by dry deposition until the snow completely melts."

- *Line 506-509 add necessary references to support your discussions.*

**Response:** Thank the reviewer for this suggestion. Based on the previous comments and the inclusion of the scatter plot, we have accordingly revised the discussion, and the original content from Lines 506-509 has been removed. However, the relevant points have been addressed in the conclusion, where we have added the necessary references related to the content from the original Lines 506-509.

- *Suggest to add more quantitative bias analysis, especially for Section 3.4.2 Nitrate concentrations and spatial distribution.*

**Response:** We thank the reviewer for the suggestion. Another reviewer had a similar comment. Now we have added specific quantitative values to the bias analysis in Section 3.4.2 as "In particular, in high-pollution regions like Jilin Province, the model exhibited a negative bias, with an average observation-to-simulation ratio of 1.7, corresponding to a Normalized Mean Bias (NMB) of 40.29%."

- *Line 543-549 add necessary references to support your discussions.*

**Response:** We thank the reviewer for this suggestion. We have incorporated necessary references and, based on previous comments and the inclusion of the scatter plot, we have accordingly revised the discussion. The new statement of this part is as follows:

**From Page 23, Line 543 in the original manuscript:** "The most likely reason for the discrepancies in NITS between the model and observations is the underestimation of atmospheric nitrate concentrations, which probably originates from the model's underestimate of $NO_x$ emissions in this region. Additionally, uncertainties in the deposition processes (Akter et al., 2023; Huang et al., 2015; Lu and Tian, 2014), including dry and wet deposition of nitrate from the atmosphere to the snowpack, could also play a role. Furthermore, post-depositional processes

could further contribute to the differences between the model and observations. These processes include snowfall dynamics, snow accumulation, and gas and aerosol scavenging in the snow (An et al., 2022; Flanner et al., 2012; Li et al., 2022; Poschlod and Daloz, 2024; Qian et al., 2014; Zhao et al., 2014), all of which may introduce uncertainties in the simulation of NITS. Another factor contributing to these discrepancies could be the relatively coarse model resolution, as it may not sufficiently capture the heterogeneous spatial distributions of snow and nitrate concentrations, especially when fine-scale variations are significant (Berg et al., 2024; Yu, 2013)."

- ***Line 516-519 you mentioned comparing simulated monthly values with observed daily values should be cautioned due to the significant temporal fluctuations in NITS, why did you do daily-to-daily comparisons as you can output daily results.***

**Response:** Thanks for your comment. Here we mentioned it is not a comparison of model monthly values versus observed daily values. Although snow samples are collected on specific dates, they do not represent only the conditions of that day. Snow accumulation is a cumulative process, and the data collected reflects the conditions accumulated over a certain period prior to the sampling date. The challenges mentioned here arise from the difficulty in determining whether an observation accurately represents conditions from the past few days, weeks, or even a month. Indeed, while the model can output daily results, the observed data does not represent daily conditions. Instead, the samples reflect an accumulation of conditions over the past few days or weeks.

- ***Line 543-549 is repeated by Line 555-563 more or less, please simplify.***

**Response:** We sincerely thank the reviewer for careful review and valuable suggestions. We have now deleted and merged the redundant content in the conclusion section. Also, we have incorporated necessary references and, based on previous comments and the inclusion of the scatter plot, we have accordingly revised the discussion. The new statement of this part is as follows:

[revised manuscript text omitted]

---

## Author Response (AR2)

**Remarks from the preceding review file validation**

*Figures 1,2, 3, 4, 5, 6, 7, 8, 9 and S2 may contain a territory that is disputed according to the United Nations. If and when the manuscript is accepted for final revised publication, you will be asked to choose one of the following options: (a) you could remove the disputed territory from the map and submit new figure files, or (b) we could add a statement that some figures contain disputed territories.*

**Response:** Thanks for the reminder. After careful consideration, we would like to proceed with the second option, i.e., to add a statement indicating that some figures contain disputed territories.